# The Effect of Different Water Extracts from *Platycodon grandiflorum* on Selected Factors Associated with Pathogenesis of Chronic Bronchitis in Rats

**DOI:** 10.3390/molecules25215020

**Published:** 2020-10-29

**Authors:** Waldemar Buchwald, Michał Szulc, Justyna Baraniak, Natalia Derebecka, Małgorzata Kania-Dobrowolska, Anna Piasecka, Anna Bogacz, Monika Karasiewicz, Joanna Bartkowiak-Wieczorek, Radosław Kujawski, Agnieszka Gryszczyńska, Piotr Kachlicki, Mariola Dreger, Marcin Ożarowski, Anna Krajewska-Patan, Małgorzata Górska-Paukszta, Ewa Kamińska, Przemysław Ł. Mikołajczak

**Affiliations:** 1Department of Botany, Breeding and Agricultural Technology of Medicinal Plants, Institute of Natural Fibres and Medicinal Plants, Kolejowa 2, 62-064 Plewiska, Poland; anna.patan@iwnirz.pl; 2Department of Pharmacology, Poznań University of Medical Sciences, Rokietnicka 5a, 60-806 Poznań, Poland; mszulc@ump.edu.pl (M.S.); radkuj@ump.edu.pl (R.K.); kaminawe@gmail.com (E.K.); przemmik@ump.edu.pl (P.Ł.M.); 3Department of Pharmacology and Phytochemistry, Institute of Natural Fibres and Medicinal Plants, Kolejowa 2, 62-064 Plewiska, Poland; justyna.baraniak@iwnirz.pl (J.B.); malgorzata.kania@iwnirz.pl (M.K.-D.); aniabogacz23@o2.pl (A.B.); agnieszka.gryszczynska@iwnirz.pl (A.G.); 4Laboratory of High Throughput Technologies, Institute of Molecular Biology and Biotechnology, Faclty of Biology, Adam Mickiewicz University, Umultowska 89, 61-614 Poznań, Poland; nataliad@amu.edu.pl; 5Department of Pathogen Genetics and Plant Resistance, Metabolomics Team, Institute of Plant Genetics, Polish Academy of Sciences, Strzeszyńska 34, 60-479 Poznań, Poland; apiasecka@ibch.poznan.pl (A.P.); pkac@igr.poznan.pl (P.K.); 6Institute of Bioorganic Chemistry, Polish Academy of Sciences, Noskowskiego 12/14, 61-704 Poznań, Poland; 7Laboratory of International Health, Department of Preventive Medicine, Poznań University of Medical Sciences, Święcickiego 6, 60-781 Poznań, Poland; mkarasiewicz@ump.edu.pl; 8Department of Physiology, Poznań University of Medical Sciences, Święcickiego 6, 60-781 Poznań, Poland; joanna@wieczorek.net.pl; 9Department of Biotechnology, Institute of Natural Fibres and Medicinal Plants, Wojska Polskiego 71b, 60-630 Poznań, Poland; mariola.dreger@iwnirz.pl (M.D.); marcin.ozarowski@iwnirz.pl (M.O.); malgorzata.gorska-paukszta@iwnirz.pl (M.G.-P.)

**Keywords:** *Platycodon grandiflorum*, extracts, bronchitis, mRNA expression, cytokines, saponins

## Abstract

The aim of this study was to assess the activity of extracts from *Platycodon grandiflorum* A. DC (*PG*) in a model of chronic bronchitis in rats. The research was carried out on three water extracts: E1 – from roots of field cultivated *PG*; E2 – from biotransformed roots of *PG*; E3 – from callus of *PG.* The extracts differed in saponins and inulin levels—the highest was measured in E3 and the lowest in E1. Identification of secondary metabolites was performed using two complementary LC-MS systems. Chronic bronchitis was induced by sodium metabisulfite (a source of SO_2_). Animals were treated with extracts for three weeks (100 mg/kg, intragastrically) and endothelial growth factor (VEGF), transforming growth factors (TGF-β1, -β2, -β3), and mucin 5AC (MUC5AC) levels were determined in bronchoalveolar lavage fluid, whereas C reactive protein (CRP) level was measured in serum. Moreover, mRNA expression were assessed in bronchi and lungs. In SO_2_-exposed rats, an elevation of the CRP, TGF-β1, TGF-β2, VEGF, and mucin was found, but the extracts’ administration mostly reversed this phenomenon, leading to control values. The results showed a strong anti-inflammatory effect of the extracts from *PG*.

## 1. Introduction

The features of different respiratory tract infections depend strongly on the structures in which the inflammatory process is observed. Infections occurring in this tract are classified as upper and lower respiratory tract infections, which exist in the trachea, bronchial tubes, the bronchioles, and the lungs. Lower respiratory tract infection should always be taken seriously since it is very likely to cause serious morbidity or even death. The two most dangerous infections of the lower respiratory tract are bronchitis and pneumonia. Symptoms of chronic bronchitis include long-lasting cough and increased mucus production in the lungs. Mucus hypersecretion is a characteristic feature not only of chronic bronchitis but also of several different human airway diseases, including cystic fibrosis and asthma [1]. Other characteristic symptoms of bronchitis are wheezing, and shortness of breath. Chronic bronchitis is based on a constant inflammatory process in the lining of the bronchial tubes. The inflammatory process is sustained by a few pro-inflammatory cytokines: interleukin 6 (IL-6) and transforming growth factor TGF-β [2]. A significant increase in the levels of TGF-β1 in bronchoalveolar lavage (BALF) fluid was also found in patients with chronic idiopathic cough [3].

The extract from *Platycodon grandiflorum* A. DC (*PG*) root, widely used in traditional medicine in Asia as a remedy for bronchitis, asthma, pulmonary tuberculosis, and inflammatory diseases, may be a very promising option in the treatment of chronic bronchitis. It seems that the use of root of large-flowered *PG* could replace the discontinued traditional raw materials used in the treatment of respiratory diseases: comfrey root (*Symphytum officinale* L.), coltsfoot leaf and flower (*Tussilago farfara* L.) due to the presence of harmful pyrrolizidine alkaloids [4].

In traditional Chinese medicine, this plant is regarded to be an expectorant and antitussive agent in the treatment of cough, cold, and tonsillitis. Moreover, in Korea, it is known to be useful in the treatment of bronchitis, asthma, pulmonary tuberculosis, and inflammatory diseases [5]. *PG* belongs to the bellflower family (*Campanulaceae*) native to Asia and widely distributed in Northern Asia, People’s Republic of China, the Democratic People’s Republic of Korea, Japan, the Republic of Korea, and the Russian Federation. Other common names of this plant are Chinese bellflower, Japanese bellflower, and platycodon. The plant material of the dried root is of particular laboratory and medicinal interest. The *PG* roots are reported to contain carbohydrates, proteins, lipids, as well as small amounts of phenolics [6]. A phytochemical investigation revealed that this product also contains flavonoids, polyacetylenes, sterols, and other bioactive compounds [7]. However, the major chemical constituents with expectorant and anti-inflammatory activities [8,9,10,11,12,13] found in the roots of *PG* are triterpene saponins named platycodins (A, D, D2, and D3), 2′′- and 3′′-*O*-acetyl polygalacin D2, platyconic acid A, and platycosides [7,8]. The *PG* extracts and some of major components of the plant, such as platycodin D and platycodin D3, together with anti-inflammatory activity have been found to have other pharmacological activities, including anti-allergic activity [14], the ability to augment immune responses [15], the ability to stimulate apoptosis in skin cells [16], exert beneficial effects, suggesting that they can be a potentially effective therapeutic agent for atopic dermatitis [17], anti-obesity, and hyperlipidemia effects [18] and a protective effect against oxidative hepatotoxicity [6]. Platycodin D and D3, isolated from the root of *PG,* increased the secretion of TNF-α as well as expression of TNF-α mRNA in activated RAW 264.7 cells [19]. 

Mucin is a glycoprotein expressed in respiratory epithelial cells. It is the main component of mucus that protects the respiratory tract from infection and toxins [1,20], but its overproduction occurs in respiratory diseases such as chronic bronchitis, produces sputum, and leads to disease progression [21]. Concerning the role of platycodin D and D3 from *PG,* it was found that these compounds increase mucin release from rat and hamster tracheal surface epithelial cell cultures and also from intact rat trachea upon nebulization [11]. Moreover, aqueous extract of the root of *PG* stimulated the secretion of airway mucin in a sulfur dioxide-induced bronchitis in rat model. This extract regulated the production and secretion of airway mucin and these results may explain the traditional use of aqueous extract of *PG* as an expectorant [22].

It is commonly known that the obtained plant raw material cultivated traditionally is often characterized by a large variation in the content of active substances (plant secondary metabolites), which is affected by the conditions of growing of medicinal plants (e.g., temperature, photoperiod, amount of rainfall, type of soil) [23]. In contrast, in vitro techniques provide a perfect opportunity to produce a large amount of plant media biomass, rich in bioactive compounds, without environmental restrictions. For example, biotechnological methods that allow to receive a stable amount of active substances, often increasing their content for years, have also been used for *PG*, e.g., elution with biotic stressors (yeast, methyl jasmonate, salicylic acid) and genetic transformation (strain *A. rhizogenes*, transformed root cultures) [24]. Transformed roots are characterized by genetic and biochemical stability enabling breeding for many years, while maintaining a high level of secondary metabolite production [25]. Therefore, in this study, the subject of research was material in the form of extracts obtained from *PG* cultivation by three methods: from roots of field-cultivated plant (E1), from its biotransformed roots (E2), and from callus of *PG* (E3). 

The aim of this study was to assess the pharmacological activity of these three extracts from *PG,* differing in the composition of active compounds in a model of chronic bronchitis in rats. Effects of the extracts on concentrations of several factors (cytokine-like factors-more or -less related to inflammatory processes) involved in pathogenesis of chronic bronchitis (vascular endothelial growth factor (VEGF), transforming growth factors (TGF-β1, TGF-β2, TGF-β3), mucin 5AC (MUC5AC), and C-reactive protein (CRP) were determined and their mRNA levels were also assessed. This knowledge may be particularly important for understanding the role of herbal products from *PG* in their possible application for the treatment of chronic bronchitis.

## 2. Results

### 2.1. Inulin and Saponin Contents 

Inulin and saponin (sum of saponins) contents were measured in the E1, E2, and E3 extracts (Table 1). It was found that the sum of saponins concentration differed in the extracts (ANOVA F(2,9) = 440.5, *p* ≤ 0.001). Moreover, the highest concentration of saponins was observed in E3, whereas in E1, the lowest content was observed. Similarly, a one-way ANOVA revealed significant differences in inulin contents in all extracts (ANOVA F(2,9) = 31.64, *p* ≤ 0.001) and the highest and lowest concentrations of inulin were found in E3 and E1, respectively.

### 2.2. Qualitative Analysis 

The HPLC-MS^n^ and UPLC-MS/MS analysis (Figure 1, Figure 2 and Figure 3) enabled the identification of 48 different compounds (Table 2), mainly oleanane-type triterpenoid saponins (23 structures). 

Secondary metabolites from phenylpropanoid pathways were also present. The *O*- and *C*-glycosides of flavones—apigenin, chrysoeriol, and luteolin—as well as glycosides of the flavonol quercetin were also identified. In addition, caffeic acid derivatives and glucosides of lignan lariciresinol were also observed. A great diversity of glycosidic substitution of phenylpropanoid structures was identified in *PG* roots. The MS analysis in negative ionization modes enabled 7-O-, 6-*C*- and 2′′-*O*- glycosidic bonds to be distinguished as well as 6′′-acylation with ferulic and sinapic acids (from the hydroxycinnamic acid group) in flavonoids, as previously described by Piasecka et al. [26]. Compounds No. 1–5 and 13 were identified as glycosidic derivatives of caffeic acid (also from hydroxycinnamic acids group) substituted with glycosidic and quinic moieties. 

Triterpenoid saponins constitute a large part of the detected metabolites (metabolites No. 15, 17, 19–23, 25, 27, 30, 31, 33–38, 41–46, 48). All of these compounds were tentatively identified according to the literature referred in Table 2 on the basis of exact mass measurement followed by chemical formula calculation and the fragmentation scheme. The structures of the saponins, in particular the places of substitution with glycosidic and acyl moieties, as well as character of the substituents, could not be precisely defined by mass spectrometry and require further detailed NMR analysis.

The LC-MS profiles of secondary metabolites varied among samples significantly. The highest number of metabolites (27) was identified in extract E2, while 19 and 17 metabolites were detected in extracts E1 and E3, respectively (Figure 4, Table 3). 

Only one compound, No. 19, (platycodigenin gentobioside) was observed in all studied samples. Extracts E2 and E3 had the highest number of common structures. Only three compounds were common for E1 and E2 (No. 23, 45 and 48). Nevertheless, all of the metabolites that were observed in two or three extracts belonged to saponins. Phenylpropanoids were highly diverse among samples and were mainly tissue specific. Caffeic acid glycosides and 3-caffeoylquinic acid were characteristic only for extract E2, whereas the isomer 5-caffeoylquinic acid and its glycoside were detected in extract E3. Flavonoid glycosides were identified mainly in extract E1 (No. 6, 7, 9, 14, 24, 28, 29, 32, 40, 47). Therefore, the differences in pharmacological effects among extracts can be related with diversification of those class of metabolites.

### 2.3. Cytokine-like Factors’ Concentration Changes

In rats with chronic bronchitis induced by MBS concentrations of selected factors VEGF, TGF-β1, TGF-β2, TGF-β3, MUC5AC, and CRP were determined using appropriate ELISA methods. 

It was found that animals exposed to SO_2_ produced increased levels of CRP (ANOVA F(4,41) = 684.5, *p ≤* 0.001), but E1 and E2 treatment significantly lowered (by 33% and 44%, respectively) this protein level, suggesting that these extracts showed anti-inflammatory activity (Figure 5). 

Moreover, the SO_2_-exposed rats showed significant elevation of TGF-β1 (ANOVA F(4,41) = 23.6, *p* ≤ 0.001), TGF-β2 (ANOVA F(4,41) = 15.0, *p* ≤ 0.001), VEGF (ANOVA F(4,41) = 20.2, *p* ≤ 0.001) level and mucin (ANOVA F(4,41) = 40.6, *p* ≤ 0.001) concentration in BALF (Table 3). 

The effect of water extracts from *PG* on TGF-β1 and TGF-β2 concentrations in BALF is not clear. The E1, E 2, E3 extracts decreased the concentration of TGF-β1 in SO_2_-exposed rats, whereas no effect of extracts on TGF-β2 was observed (Table 3). Results obtained for TGF-β3 were unclear, because all values obtained for this factor were lower than the sensitivity detection level (data not shown). In SO_2_-exposed animals, an increase of VEGF concentration in BALF was detected, whereas all three extracts with the same strength lowered this parameter in more or less the same way (Table 3). Similarly, SO_2_-exposed animals showed an increase of mucin concentration in BALF, whereas all extracts decreased the concentration of this protein (Table 3). 

### 2.4. Selected Genes’ mRNA Level Changes

In general, control-SO_2_-exposed rats showed a significant elevation of TGF-β1 (total variation ANOVA F(4,41) = 15.1, *p* ≤ 0.001), TGF-β2 (total variation ANOVA F(4,41) = 167.3, *p* ≤ 0.001) and VEGF (total variation ANOVA F(4,41) = 43.2, *p* ≤ 0.001) mRNA expression in bronchi (Table 4), while the decrease of mucin mRNA expression (total variation ANOVA F(4,41) = 28.9, *p* ≤ 0.001) was observed (Table 4). The strongest elevation of mRNA VEGF expression (200-fold increase) was detected in control SO_2_-exposed rats, when taking into consideration the results of the VEGF factor. Overall, statistically significant variability in results of TGF-β3 mRNA was also calculated (ANOVA F(4,41) = 22.1, *p* ≤ 0.001) in bronchi, however, there was no statistical difference between control+SO_2_ and not-SO_2_-exposed control rats (Table 4).

Generally, all extracts significantly lowered the mRNA level of VEGF, TGF-β1, and TGF-β2 in the bronchi (Table 4), thus, reversing the effect of SO_2_ exposure. In turn, for mucin mRNA, the administered extracts increased expression to values even higher than those observed in control animals (control group). The exact effect of the extracts on mRNA of TGF-β3 is unclear, because SO_2_ exposure alone did not affect expression, whereas administration of extracts (especially E2 and E3), increased the expression to values even higher than those observed for control animals (control group) (Table 4). 

In lungs of control SO_2_-exposed rats, mRNA expression generally showed a significant elevation of TGF-β1 (total variation ANOVA F(4,41) = 52.9, *p* ≤ 0.001), TGF-β2 (total variation ANOVA F(4,41) = 261.3, *p* ≤ 0.001) and mucin (total variation ANOVA F(4,41) = 28.9, *p* ≤ 0.001) (Table 5), while simultaneously the decrease of TGF-β3 mRNA expression (total variation ANOVA F(4,41) = 73.7, *p* ≤ 0.001) was observed.

Treatment with all extracts caused a decrease of mRNA expression both in mRNA of TGF-β1 and TGF-β2 in the lungs (Table 5), and the strongest effects after E2 and E3 administration were shown. On the contrary, only the administration of E1 extract led to the normalization of TGF-β3 mRNA expression. In turn, considering the mRNA mucin expression results in lungs, it was noted that only the E2 extract showed a significant statistical decrease towards the values in control group, whereas E1 and E3 administration produced higher mRNA expression than control SO_2_-exposed rats (Table 5).

## 3. Discussion

Chronic cough and other issues of respiratory tract diseases are a serious clinical problem. Inflammatory reactions in the bronchial mucosa in respiratory tract diseases are controlled by set of interacting factors, i.e. cytokines. To explore the possible role of airway wall remodeling in chronic cough, some experimenters measured the levels of growth factors, such as TGF-β in BALF fluid, and examined the expression of TGF-β in airway submucosa of chronic idiopathic cough patients [3]. Hypersecretion of airway mucus, which contributes to the pathophysiology of numerous respiratory diseases, was also examined [36]. Moreover, platycodin D, a triterpenoid saponin isolated from the root of *PG*, was found to be an anti-inflammatory factor in a mouse model of allergic asthma [37].

In this study, we focused on the effects of the extracts from *PG* on the concentrations of a few factors possibly involved in pathogenesis of chronic bronchitis (VEGF, TGF-β1, TGF-β2, TGF-β3, MUC5AC, and CRP). 

Our results showed that extracts from roots of field cultivated *PG* (E1) and extracts from biotransformed roots of *PG* (E2) treatment significantly lowered CRP concentration (Figure 5), suggesting that these extracts showed anti-inflammatory activity. This is in line with previous literature, which clearly demonstrated that animals exposed to SO_2_ develop chronic inflammatory processes that are sustained by pro-inflammatory mediators, mainly cytokines. Intensified expression of factors strongly connected with inflammation is also observed [8,9,38]. For example, Choi et al. [9] investigated the inhibitory effect of aqueous extracts from the roots of *PG* (Changkil: CK) on airway inflammation in a murine model of asthma. Extracts from the roots of *PG* markedly decreased the number of infiltrated inflammatory cells and the levels of Th1 and Th2 cytokines and chemokines compared with the control group and reduced ovalbumin-specific IgE levels in BALF fluid [9]. Choi et al. [9] also investigated the inhibitory effects of *PG* root-derived saponins on ovalbumin-induced airway inflammation in mice. Derived saponins suppressed the number of leukocytes, IgE, Th1/Th2 cytokines, and MCP-1 chemokine secretion in BALF fluid. Additionally, ovalbumin-increased MUC5AC mRNA expression. The NF-κB activation, leukocyte recruitment, and mucus secretion were inhibited by saponins from *PG*. Therefore, they demonstrated the anti-inflammatory activity of saponins from roots of *PG* [39]. Additionally, researchers focused on the anti-inflammatory mechanisms of eight platycodin saponins isolated from the roots of *PG*. Results obtained from their investigations suggest that the main inhibitory mechanism of platycodin saponins may be the reduction of iNOS and COX-2 gene expression through blocking of NF-κB activation [8].

However, neither exact mechanisms of action of platycodin saponins nor their effects on the production of proinflammatory mediators in cells are fully understood, which is reflected in our results, since the ability of E1 and E2 extracts to decrease the CRP level but not E3 shows that the inflammatory process caused by SO_2_ exposure was lowered, but this meant that it was not in a simple correlation with the content of saponins or inulin (Table 1). It does not match with the results of other authors suggesting that saponins isolated from *PG* root can significantly inhibit the excessive production of NO, PGE2 and pro-inflammatory cytokines, including interleukin-1β (IL-1β) and TNF-α in a concentration-dependent manner without causing any cytotoxic effects [13]. However, it should be emphasized that the cited results were obtained from in vitro studies, while our results came from an in vivo experiment.

The differences obtained in this study are not only related to different content of saponins and inulin (Table 1), but also to a slightly different qualitative composition of the examined extracts, as mentioned in the Results section (Chapter 2.2., Qualitative Analysis). This may be in line with the observation that only one compound, No. 19, (platycodigenin gentobioside) is common to all extracts (Figure 4). In addition to saponins (No. 15, 17, 19, 20, 21, 22, 23, 25, 26, 27, 30, 31, 33, 34, 35, 36, 37, 38, 39, 41, 42, 43, 44, 45, 46, 48), the presence of other groups of chemical compounds can be noticed (flavonoids: No. 6, 7, 9, 14, 24, 28, 29, 32, 40, 47; hydroxycinnamic acids: No. 1, 2, 3, 4, 5, 11, 16; lignans: No. 8, 10, saccharide: No. 12), which varied significantly among samples from the extracts (for example: flavonoids only in E1 extract). This is consistent with the knowledge of the possible contribution of these compounds present in PG to various pharmacological effects on inflammatory basis [40]. Hence, it cannot be ruled out that the observed differences in anti-inflammatory activity may have been caused by the presence of flavonoid glycosides, which were identified mainly in the E1 extract, whereas caffeic acid glycosides and 3-caffeoylquinic acid were characteristic only for E2 extract, and the isomer 5-caffeoylquinic acid and its glycoside were detected in the E3 extract (Table 3). Nevertheless, their possible significance should be supported by detailed quantification (content) of the above-mentioned compounds in the future.

In our study, elevation of TGF-β1 and TGF-β2 in control SO_2_-exposed rats was correlated with the increase of mRNA expression of TGF-β2 in lungs and bronchi (Table 3, Table 4 and Table 5). It has been established that TGF-β is one of the cytokines with expression in macrophages and epithelial tissue. The isoform TGF-β1 is responsible for fibrinogen activity in chronic bronchitis and plays a role in airway wall remodeling [2]. Moreover, this isoform produced the inhibitory action on immune cell differentiation (Th1 and Th2 cells and B cells) and cytokine production (IFN-γ and IL-2) [41]. Xie et al. [3] observed a significant increase in levels of TGF-β in BALF, which suggests that TGF-β may be important in the airway remodeling changes observed in chronic idiopathic cough patients [3]. Taking this into consideration, this situation indicates a clear connection between SO_2_ exposure and levels of TGF-β1 and TGF-β2 in the chronic bronchitis model.

In this study, the effect of extracts from *PG* on TGF-β1 and TGF-β2 concentration in BALF is not quite clear. The extracts E1, E2, and E3 lowered TGF-β1 concentration in SO_2_-exposed rats, and the effect of E3 was the strongest, but this meant that it was not a simple correlation with contents of saponins or inulin (Table 1 and Table 3). A similar relationship was observed at the level of mRNA in the lungs of animals (Table 5), whereas in bronchi, all extracts reduced mRNA to the same level (Table 4). In turn, for TGF-β2, no effect of the extracts in BALF was observed, but expression at the mRNA level in both bronchi and lungs of animals was significantly reduced (Table 4 and Table 5) and the strongest effect was measured in lungs where E2 and E3 extracts inhibited transcript formation more than 20 times, as compared with control-SO_2_-exposed animals. This suggests that extracts containing a higher content of saponins and insulin have stronger effects.

It is known that TGF-β3 shares 80% amino acid homology with TGF-β1 and TGF-β2 [42]. However, the results obtained for this factor are unclear and different from those obtained for TGF-β1 and TGF-β2, because SO_2_ exposure induced a reduction in lung expression mRNA, while only the E1 extract increased the value of this parameter to those observed in control animals (Table 5). In contrast, SO_2_ exposure lowered the TGF-β3 mRNA in bronchi insignificantly, whereas 2 and E3 administration induced higher expression than that obtained in control animals (Table 4). Similar lowering of TGF-β3, but on a protein level, was observed in patients with severe chronic obstructive pulmonary disease (COPD) [43]. 

The differences observed in the response of TGF-β isoforms to the action of SO_2_ and the administration of extracts may be related to the slightly different distribution and action of those in airways, since TGF-β1 colocalizes with extracellular matrix proteins, such as collagen, and interfaces between epithelial and mesenchymal cells, TGF-β2 is found in endodermal bronchiolar epithelium, whereas TGF-β3 is expressed in tracheal mesenchyme and the endodermal epithelial cells in bronchioles and mesodermal cells [44]. Moreover, the biological effects of different TGF-β isoforms depend on their availability, combination of two types of their receptors, and intracellular signaling pathways that they can induce [42]. For example, in the airways of humans with asthma, TGF-β1 levels are elevated as compared with normal control subjects, suggesting a role in the repair of injured asthmatic airways or the existence of a negative feedback loop controlling airway inflammation [44]. Similarly, an increase in TGF-β2 expression in asthmatic epithelium was shown, which correlates with an increase in the number of eosinophils and neutrophils in patients with severe and mild asthma [42]. There is relatively little information on TGF-β3 expression, although the available data suggests that there is no difference in TGF-β3 expression between asthmatic patients and control subjects [42], but in patients with severe COPD, the lowering of TGF-β3 was found [43]. In conclusion, SO_2_, as a pro-inflammatory factor, induced changes in the expression of individual TGF-β isoforms in a similar way as the above-mentioned. However, the result of administration of the extracts to individual isoforms is not easily explained and will likely, as was mentioned previously, require in-depth research on the effect of not only saponins and inulin occurring in various amounts in the studied extracts, but also compounds from other chemical groups. 

High-molecular-weight mucous glycoproteins, named mucins, are produced by goblet cells in the epithelium and sero-mucous glands in the submucosa. The MUC5AC and MUC5B glycoproteins, located on chromosome 11p15.5, are considered to be the major mucins both in normal respiratory tract secretions and airway secretions in patients with respiratory tract diseases. Mucus is a viscoelastic gel that forms a thin film on the surface of the airways and protects the epithelial lining against bacteria and viruses, and clears them from the airway by ciliary movement [36]. It is notable that overexpression of mucin is common in various disorders of the respiratory tract such as asthma, COPD, chronic bronchitis, and cystic fibrosis [36,45]. Many different inflammatory mediators also increase mucin gene expression and mucin synthesis in cells of the respiratory tract.

We found the overexpression of mucin in BALF in control-SO_2_-exposed rats (Table 3). It is in line with the results of experiments of other authors. They also detected a high expression of mucin in similar experimental conditions [46]. At the same time, we observed a lowering of the protein concentration caused by activity of all examined extracts (Table 3). In lungs, SO_2_ exposure increased a mucin mRNA expression level, whereas only E2 extract lowered this parameter towards control values (control group) (Table 5). It is possible that inflammation and also oxidative stress can induce mucin gene expression leading to mucin overproduction. Taking into consideration the obtained results concerning mucus level, we suppose that SO_2_ plays a role as a mucolytic agent in this case. However, in the bronchi, we observed a lowering of mucin mRNA expression as a result of SO_2_ activity, whereas all examined extracts increased this parameter, and the values obtained were even higher than those obtained for control animals (Table 4). This kind of discrepancy between effects in BALF, lungs, and bronchi is not easy to explain, although a similar phenomenon was observed in a small group of patients with bronchiolitis obliterans syndrome where the mRNA expression for costimulatory molecules B7-1 (CD80), and the B7-1/GAPDH and B7-2/GAPDH ratios were significantly elevated for bronchial epithelial cells, whereas no differences in BALF were observed [47].

Therefore, even though the effects of *PG* extracts activities are not clear, it is known that their components are probably involved in regulation of mucin expression and production. Nevertheless, the results of Wang et al. provided evidence that platycodin D inhibits IL-13-induced expression of inflammatory cytokines and mucus in nasal epithelial cells by inhibiting the activation of NF-κB and MAPK signaling pathways [37]. The results obtained by Ryu et al. [22] suggest that aqueous extracts of *PG*, platycodin D, and deapi-platycodin can regulate the production and secretion of airway mucin. In their paper, the effects of *PG* on oversecretion of pulmonary mucin (MUC5AC) in SO_2_-induced bronchitis in rats were examined. They found that the extract stimulated the secretion of mucin in this model and they suggested that it can explain the traditional use of balloon flower extracts as expectorants to control diseases accompanied by hypersecretion of mucus [22]. Hence, based on the cited papers and our results, it can be assumed that the effect of administering *PG* extracts on the production of mucin in chronic bronchitis should be beneficial, although this issue undoubtedly needs further examination.

It is known that VEGF increases vascular permeability and is a potent mediator of angiogenesis [48], since multiple growth factors, including VEGF and angiopoietin 1, are involved in the formation of new vessels and remodeling of existing vessels [49]. Changes in the microvasculature in chronic diseases may be out of proportion to the increased metabolic needs of tissues because of the overproduction of growth factors that stimulate vessel growth and remodeling. In SO_2_-exposed rats, we observed an increased VEGF level on protein in BALF and mRNA on bronchi, respectively (Table 3 and Table 4). All examined extracts possessed the ability to decrease VEGF level and mRNA expression. These changes correspond to the results obtained by other authors. For example, they showed a clear connection between chronic bronchitis and VEGF level in patients [48,50] and also in the model of chronic airway inflammation [49].

Recently, it was found that fermented *PG* extracts can relieve airway inflammation in lipopolysaccharide/ovalbumin-induced asthma mice, since these extracts significantly reduced the concentration of proinflammatory cytokine IL-17E in the BALF and the serum [51].

Generally, the obtained results are not in a simple relation with the content of saponins or inulin. However, it should be emphasized that the extracts used differ in the composition, in particular of the saponins and other analyzed compounds, e.g., phenylpropanoids, as well as in the content of saponins (Table 2, Figure 4). Thus, it can be presumed that both groups of compounds may contribute differently to the observed effects, depending on the extracts used.

Although the understanding of the pharmacological profile of the action of compounds derived from *PG* has increased [40], accurate and unambiguous determination of their contribution to the pharmacological effects described in this paper requires further detailed studies.

## 4. Materials and Methods

### 4.1. Plant Material

Plants of *PG* were identified by Prof. Jan Kozłowski. A voucher specimen was deposited at the Department of Pharmaceutical Biology and Medicinal Plant Biotechnology, Medical University, Warsaw, Poland (Accession No. FW 21/027/2004). Roots of *PG* were obtained from conventional tillage conducted at the Institute of Natural Fibres and Medicinal Plants (INF&MP), Plewiska, near Poznań, Poland. The field cultivation was established in spring by sowing seeds directly into ground (lessivé soil with granulometric composition of light loamy sands). Roots were harvested in autumn after the second year of plant growth. They were washed, cut into small pieces, and dried in a drying chamber at a temperature of 40 °C and relative humidity of 20%. 

For surface disinfection, seeds of *PG* (derived from white blooming plants obtained from the collection of Garden of Medicinal Plants, INF&MP, Plewiska) were placed in 70% alcohol for 2–3 min, then placed in Domestos^TM^ solution (1:4) for 20 min and rinsed with sterilized water three times. The seeds were cultured on MS medium (according to Murashige and Skoog [52]) without growth regulators to germinate. Callus was induced from root explants (1 cm) and cultured on MS medium supplemented with BAP (2.0 mg/L), NAA (1.0 mg/L) and adenine chloride (1.0 mg/L) containing 30 g/L of saccharide and 8 g/L agar (Bactoagar) (pH = 5.6). Callus was cultured in the dark at 23 ± 2 °C temperature and sub-cultured every four weeks. Regarding good biomass growth parameters, the callus line was chosen for the study and propagated. Callus tissue was collected, weighed, and dried at room temperature every four weeks. 

For hairy root induction in *PG*, leaf and stem tissues of micropropagated plants were infected with *Agrobacterium rhizogenes* ATCC 15834. The explants were cultured on phytohormone-free half-strength MS solid medium containing 4% sucrose in the dark for four days and then transferred to the same medium supplemented with Claforan (500 mg/L). The explants developed new roots from the wounded tissues after 15 days of culture in the light. The hairy roots were individually excised and placed on the same medium with the antibiotic mentioned above. Hairy root lines showing rapid growth were maintained on hormone-free Woody Plant Medium with sucrose (40 g/L), which has proved to be the best for optimal growth. The culture was incubated in the dark at 25 ± 2 °C on a rotary shaker (100 rpm) and sub-cultured every six weeks. The transformation was confirmed by a polymerase chain reaction (PCR) experiment. Hairy roots were collected and air-dried at a temperature of 25 °C.

### 4.2. Preparation of Plant Extracts

The objects of the research were three water extracts: E1 – extract from roots of field-cultivated *PG****,*** E2 – extract from biotransformed roots of *PG*, E3 – extract from callus of *PG.* The powdered dry roots or calluses were extracted with purified water for 3 h at 90 °C (1:10 plant material to solvent ratio). After filtering, the extracts were frozen at –55 °C and then lyophilized. The dry plant extracts were stored at a temperature of 20–25 °C [53]. All extracts were prepared in the Department of Pharmacology and Phytochemistry, INF&MP, Plewiska, Poland.

### 4.3. Saponin and Inulin Contents

#### 4.3.1. Sum of Saponins Content 

The method determines the sum of saponins in E1, E2, and E3 based on the method in the dry water extract from the root of the balloon flower (*PG*) using the precipitation-weight method after previous extraction with methanol [54,55]. Briefly, a dry extract of about 2.0 g was placed in a thimble in a Soxhlet apparatus, diluted with 50 mL of methanol and left for 15 h. Next, 50 mL of methanol was added and the mixture was extracted for 6 h. The filtrate was evaporated under reduced pressure to a volume of 15–20 mL. After cooling, 50 mL of diethyl ether was added, then the solution was mixed thoroughly and allowed to clarify. The clear supernatant solution was discarded. The precipitate was purified by adding methanol in the amount of 20, 10, and 5 mL several times, further heated and filtered. The obtained filtrates were combined and evaporated under reduced pressure to a volume of 15–20 mL. After cooling, 50 mL of ethyl ether was added and the procedure was repeated as described above. After re-purification, the methanol filtrates were placed in a pre-weighed, round-bottomed flask, then the solvent was completely stripped in a vacuum and dried to a constant weight at 105 °C.

The sum of saponins content was calculated using the formula:A [%] = (m × 100)/M(1)
where:

A—percentage of the sum of saponins in the extractM—mass of dry extract in gramsm—mass of the residue after drying to a constant mass in grams

#### 4.3.2. Inulin Content

The extraction method of inulin by Gibson et al. [56] and Kumari et al. [57] was adapted to the research. Approximately 0.1 g of powered sample of the extract was placed in a 10 mL flask. 10 mL of hot water was added, mixed, and after 5 min of incubation, the sample was filtered. Next, 2 mL of supernatant was transferred to a volumetric flask (10 mL). Afterwards, 2 mL of concentrated hydrochloric acid was added and the sample was incubated for 10 min in a boiling water bath. The sample was cooled down, then 0.5 mL of 0.5% resorcinol in 20% hydrochloric acid was added. The solution was heated for 1 min in a water bath, cooled down, and filled up with water to a volume of 10.0 mL. The absorbance of the test solution was measured at λ = 520 nm in comparison with water. 

Calibration curve—an inulin standard stock solution (1 mg/mL) was prepared and was used to prepare solutions with different concentrations. The procedure with the standard solution was the same as with the sample.

All reagents used in the experiment were of analytical grade and obtained from approved sellers.

### 4.4. Identification of Phytochemicals 

The identification of secondary metabolites present in the extracts of leaves from roots of *PG* was performed using two complementary LC-MS systems. In the first system, HPLC-DAD-MSn analysis was conducted using an Agilent 1100 HPLC instrument with a photodiode-array detector (Palo Alto, CA, USA) and Esquire 3000 ion trap mass spectrometer (Bruker Daltonics, Bremen, Germany) with the XBridge C18 column (150 × 2.1 mm, 3.5 μm particle size, Waters, Milford, MA, USA); the MSn spectra were recorded separately in the negative and positive ion modes, using a previously published approach [26,58]. The elution was conducted with water containing 0.1% formic acid (solvent A) and acetonitrile (solvent B). The gradient elution was started at 92% of A and linearly changed to 90% of A in 10 min, then to 75% of A in 30 min and to 2% of A over 10 min, followed by a return to the initial conditions and the column was re-equilibrated for 10 min. 

The second system consisted of UPLC with a photodiode-array detector PDAeλ (Acquity system, Waters, Milford, MA, USA) hyphenated to a high resolution Q-Exactive hybrid MS/MS quadrupole – Orbitrap mass spectrometer (Thermo Scientific, Waltham, MA, USA). Chromatographic profiles of metabolites were obtained using water acidified with 0.1% formic acid (solvent A) and acetonitrile (solvent B) with mobile phase flow of 0.4 mL/min on a BEH C18 column (2.1 × 150 mm, 1.7 μm particle size, Waters, Milford, MA, USA) at 40 °C. The injection volume was 10 μL. The gradient elution was started at 100% of A and linearly changed to 80% of A over 2 min, then to 70% of A over 8 min, and to 5% of A over 1 min, followed by a return to the initial conditions and re-equilibration for 2 min. UV absorbance was measured in the 230–450 nm wavelength range with a resolution of 2 nm and the data was exported with Empower 2 Chromatography Data Software (Waters, Milford, MA, USA).

Q-Exactive MS was operated in Xcalibur version 3.0.63 with the following settings: the HESI ion source voltage −3 kV or 3 kV; the sheath gas flow 30 L/min, auxiliary gas flow 13 L/min, ion source capillary temperature 250 °C, auxiliary gas heater temperature 380 °C. The MS/MS experiments were performed using collision energy 15 eV in Full MS/ddMS/MS scan modes in negative ionization.

The individual compounds were annotated via comparison of the exact molecular masses (with a mean error of less than 4 ppm), mass spectra and retention times to those of standard compounds, available online databases (PubChem, ChEBI, Metlin and KNApSAck) and literature. 

### 4.5. Rats

The experiments were performed on male, adult Wistar rats (220–250 g at the initiation of the experimental procedure, aged 8–9 weeks). The total number of animals at the beginning of the experiment was n = 50, housed at controlled room temperature (20 ± 0.2 °C) and humidity (60%) in a 12 h:12 h light-dark cycle (lights on: 7 a.m.). The healthy, pathogen free rats were obtained from Laboratory Animals Supplier (Ogrodowa 18, 05-840 Brwinów, Poland). Next, the animals were acclimatized for at least 1 week prior to use. Animals were kept in groups (3–4 rats/cage) in light plastic cages (60 cm × 40 cm × 40 cm) and given ad libitum access to standard laboratory chow (pellets – Labofeed B, Wytwórnia Pasz i Koncentratów, Kcynia, Poland, PN-ISO 9001:1996) and tap water. The study was conducted in accordance with the ethical guidelines for investigations on conscious animals (Polish governmental regulations - Animal Protection Act, Poland-Dz.U. (Journal of Laws) 05.33.289, 2005) and all the experimental protocol for the use of animal was approved (72/2009) by the Local Ethics Committee in Poznań, operating at Poznań University of Life Sciences, accredited by the National Ethics Committee for Animal Experiments, Ministry of Science and Higher Education, Warsaw, Poland.

### 4.6. Treatments and Groups

All animals were randomly divided into 5 groups. In 4 groups of rats (n = 40), chronic bronchitis was induced by sodium metabisulfite (MBS=Na_2_S_2_O_5_), as a source of SO_2_, according to the Pon et al. method [59]. The method based on the use of MBS gives the same effects as direct exposure to SO_2_, but it poses a lower risk to laboratory personnel. Animals placed in special container (50 cm × 50 cm × 50 cm, equipped with a fan) were exposed to MBS as an aerosol prepared from a solution containing 10% (*w*/*v*) Na_2_S_2_O_5_. The level of SO_2_ was monitored with GasHunter (Alter SA, Poland, Lot no: 12031024, ISO9001:2001) in a range of 0–2000 ppm, and the SO_2_ concentration was maintained at 500 ppm. The animals were exposed to MBS for 5 days a week (daily exposure to SO_2_ started at 9.00 a.m.)—1.0 h in the first week of experiment, 1.5 h in the second week, and 2 h in the third week of experiment (formed the positive study groups), and then they came back to their home plastic cages. Animals exposed to SO_2_ were treated for 1 h before SO_2_ exposition with the 3 extracts (forming three positive study groups exposed to SO_2_: E1+SO_2_, E2+SO_2_, E3+SO_2_) for 3 weeks once a day in a dose of 100 g/kg b.w., intragastrically (p.o.) from 1% extracts water solution. Control animals exposed to SO_2_ (positive control group = control+SO_2_) were treated with vehicle (appropriate volume of distilled water) in a volume 10 mL/kg b.w. (p.o.). The dose was chosen on the basis of our previous unpublished studies on the antitussive effect of *PG* root extract in guinea pigs and its anti-inflammatory effect in a murine model of asthma [9] and on airway mucin hypersecretion in rats [22]. Rats that were not exposed to MBS formed the appropriate control negative study group (n = 10). These animals were also placed in a special container (50 cm × 50 cm × 50 cm, equipped with a fan), but SO_2_ exposure was replaced with fresh normal air (sham-operated procedure). Experiments were performed at the Department of Pharmacology, Poznań University of Medical Sciences, Poland.

On the last day of the experiment, 2 h after the last administration of extracts, the rats were sacrificed by decapitation. Lungs and bronchi were quickly removed for further analysis according to Krasnowska et al. [60]. Briefly, the tissue of the dissected rat lungs together with the trachea was injected with a Hamilton syringe immediately after sacrifice placed in the trachea flushing 5 times = 5 lavages (5 mL total) with PBS buffer pH = 7.4 (Life Technologies) and bronchoalveolar lavage fluid (BALF) was collected. Approximately 0.2–0.3 mL of BALF was recovered after five lavages, which was then transferred to a 1.5 mL Eppendorf tube and then the fluid was stored at −20 °C. Rat’s blood was collected and centrifuged at 4000 rpm for 15 min and the serum was separated and stored at −80 °C for further analysis.

Results from a total of 36 animals from the positive study group were obtained (4 of SO_2_-exposed rats died), while data obtained from the 10 rats from the negative study group (control group) was also taken for analysis. Behavioral toxicity effects were generally not observed in any rats during the experiments. In the entire experiment, based on histopathological evaluation, there were no clear effects of acute bronchitis and no pulmonary fibrosis was observed in rats exposed to SO_2_ (data not shown).

### 4.7. Cytokine-Like Factors’ Quantification

Using appropriate ELISA methods, the concentrations of TGF-β (isoforms 1, 2 and 3), VEGF, and mucin were evaluated in BALF, whereas CRP was determined in serum. Concentrations of VEGF, TGF-β1, TGF-β2, TGF-β3, MUC5AC, and CRP were determined using ELISA tests. Determination of VEGF was estimated using a rat-specific kit (RayBiotech Inc., Peachtree Corners, GA, USA). The minimum detectable dose is typically no less than 2 pg/mL. Determination of TGF-β1 was estimated using a rat-specific kit (Enzo Life Science Inc., Farmingdale, NY, USA). The minimum detectable dose is typically no less than 11.3 pg/mL. Determination of TGF-β2 was estimated using a rat-specific kit (Hangzhou Eastbiopharm Co Ltd., Hangzhou, China). The minimum detectable dose is typically no less than 11.3 pg/mL. Determination of TGF-β3 was estimated using a rat-specific kit (Enzo Life Science Inc., Farmingdale, NY, USA). The minimum detectable dose is typically no less than 5.8 pg/mL. Determination of MUC5AC was estimated using a rat-specific kit (Enzo Life Science Inc., Farmingdale, NY, USA). The minimum detectable dose is typically no less than 0.115 ng/mL. Determination of CRP was estimated using a rat-specific kit (RayBiotech, Inc., Peachtree Corners, GA, USA). The minimum detectable dose is typically no less than 0.2 ng/mL. 

All measurements were performed using a SUNRISE-BASIC microplate reader (Tecan, Inc., Mannedorf, Switzerland). The Magellan software (ver. 1.1) was used for absorbance reading, standard curve-fitting, and calculation of the absolute value in specimen. All detection was performed with a wavelength of 450 nm. The results were calculated based on the absorbance of complex cytokines-antibodies and concentrations were obtained from model curves.

### 4.8. Assessment of the Selected Genes’ Expression Levels

#### 4.8.1. RNA Isolation and Reverse Transcription Reaction

Total RNA isolation was carried out using TriPure Isolation Reagent (Roche Diagnostics, Mannheim, Germany), according to the manufacturer’s protocol. The integrity of RNA was visually assessed by conventional agarose gel electrophoresis and the concentration was evaluated by measuring the absorbance at 260 and 280 nm in a spectrophotometer (BioPhotometer Eppendorf, Hamburg, Germany). RNA samples were stored at −80 °C until use. 1 µg of total RNA from all samples was used for the reverse-transcription into cDNA using the Transcriptor First Strand Synthesis Kit (Roche Diagnostics, Mannheim, Germany), according to the manufacturer’s protocol. Obtained cDNA samples were stored at −20 °C or used directly for the quantitative real-time PCR (qRT-PCR) reaction.

#### 4.8.2. Real-time PCR mRNA Quantification

The genes (VEGF, TGF-β1, TGF-β2, TGF-β3, Muc5AC) expression level was analyzed by a two-step quantitative real-time PCR (qRT-PCR) reaction in the lung and bronchi of rats, in 10 µL of reaction mixture, using relative quantification methodology with a LightCycler TM Instrument (Roche Applied Science, Mannheim, Germany) and a LightCycler Fast Start DNA Master SYBR Green I kit (Roche Applied Science, Mannheim, Germany), according to the instructions of the manufacturer. All primer sequences were designed using the Oligo 6.0 software (National Biosciences, Colorado Springs, CO, USA) and were verified by an assessment of a single PCR product on agarose gel and by a single temperature dissociation peak (melting curve analysis) of each cDNA amplification product (Table 6).

The GAPDH gene was used as a housekeeping gene (endogenous internal standard) for normalization of the qPCR reaction. For each quantified gene, standard curves were prepared from dilution of cDNA. All quantitative PCR reactions were repeated twice. The data was evaluated using Light Cycler Run 4.5 software (Roche Applied Science, Mannheim, Germany). Each PCR run included a non-template control to detect potential contamination of reagents (Table 7).

## 5. Statistical Analysis

All values were expressed as means ± SD. The statistical comparison of results was carried out using one-way analysis of variance (ANOVA) followed by Newman-Keuls test as a post-hoc test for detailed data analysis. The values of *p* < 0.05 were considered to indicate a statistically significant difference.

## 6. Conclusions

In conclusion, the results presented in this study suggest that all examined extracts from *PG* produce anti-inflammatory activity in male Wistar rats with chronic bronchitis induced by sodium metabisulfite. The precise mechanisms that are involved in the anti-inflammatory response of three different water extracts of *PG* are ambiguous and not completely known. Thus, all extracts could be regarded as potential alternative therapeutic factors for the prevention of inflammatory-related diseases of the respiratory tract. However, further detailed studies are still needed.

## Figures and Tables

**Figure 1 molecules-25-05020-f001:**
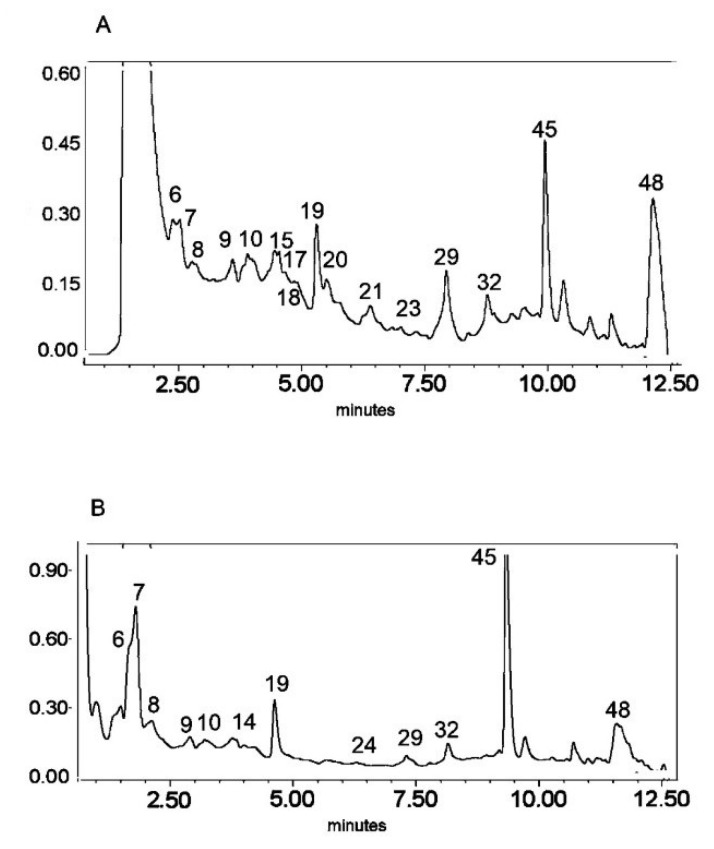
UV chromatogram of the E1 extract from *Platycodon grandiflorum* A. DC recorded at λ = 230 nm (**A**) and λ = 280 nm (**B**) by UPLC system. Chromatographic profiles of metabolites were obtained with gradients of two solvents: **A** (water acidified with 0.1% formic acid) and **B** (100% acetonitrile), mobile phase flow of 0.4 mL/min on a RP C18 column at 40 °C. The gradient elution: 100% of A linearly changed to 80% of A over 2 min, then to 70% of A over 8 min and to 5% of A over 1 min. The number corresponded to numbers of identified metabolites in Table 2.

**Figure 2 molecules-25-05020-f002:**
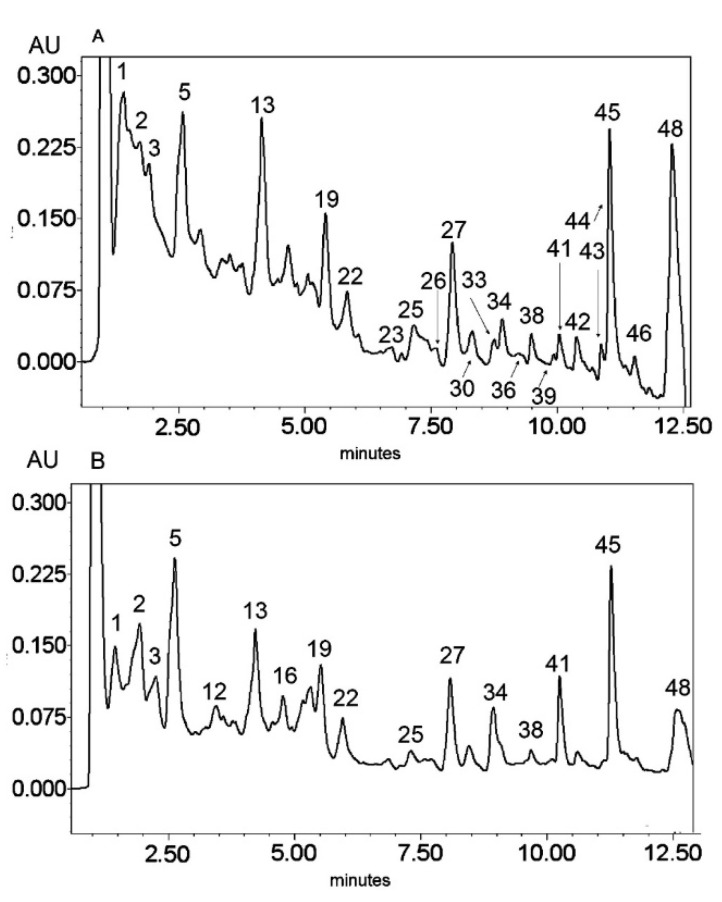
UV chromatogram of the E2 extract from *Platycodon grandiflorum* A. DC recorded at λ = 230 nm (**A**) and λ = 280 nm (**B**) by UPLC system. Chromatographic profiles of metabolites were obtained with gradients of two solvents: **A** (water acidified with 0.1% formic acid) and **B** (100% acetonitrile), mobile phase flow of 0.4 mL/min on a RP C18 column at 40 °C. The gradient elution: 100% of A linearly changed to 80% of A over 2 min, then to 70% of A over 8 min and to 5% of A over 1 min. The number corresponded to numbers of identified metabolites in Table 2.

**Figure 3 molecules-25-05020-f003:**
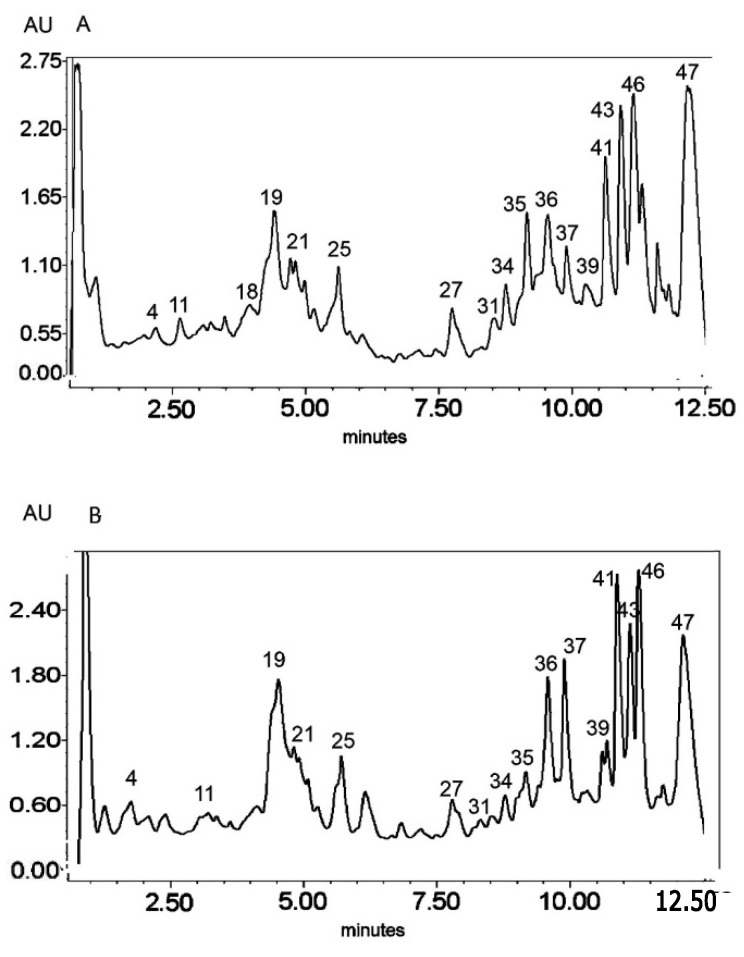
UV chromatogram of the E3 extract from *Platycodon grandiflorum* A. DC recorded at λ = 230 nm (**A**) nm and λ = 280 nm (**B**) by UPLC system. Chromatographic profiles of metabolites were obtained with gradients of two solvents: **A** (water acidified with 0.1% formic acid) and **B** (100% acetonitrile), mobile phase flow of 0.4 mL/min on a RP C18 column at 40 °C. The gradient elution: 100% of A linearly changed to 80% of A over 2 min, then to 70% of A over 8 min and to 5% of A over 1 min. The number corresponded to numbers of identified metabolites in Table 2.

**Figure 4 molecules-25-05020-f004:**
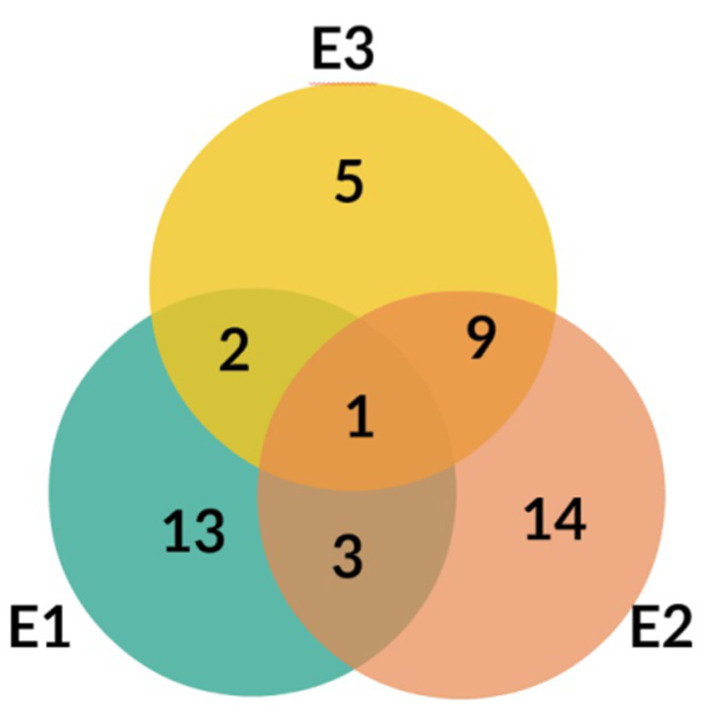
Venn diagram for E1, E2, and E3 extracts from *Platycodon grandiflorum* A. DC.

**Figure 5 molecules-25-05020-f005:**
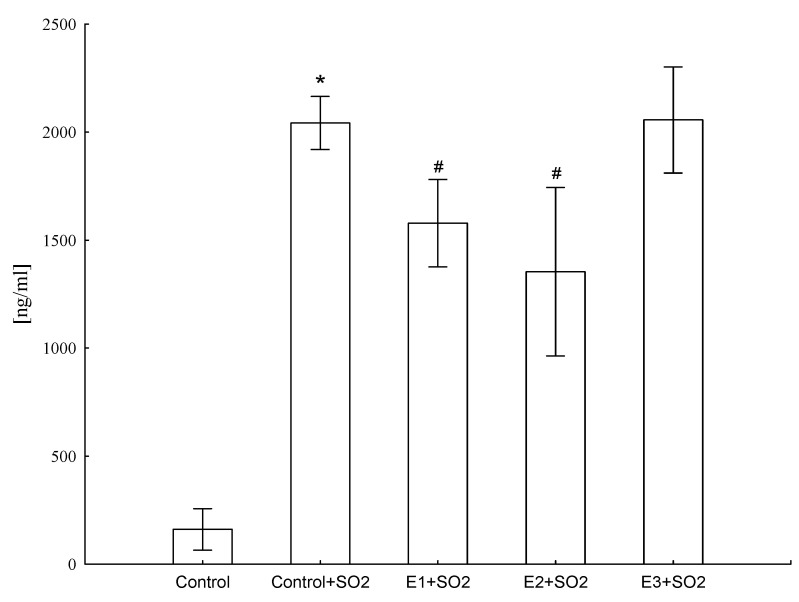
Influence of different extracts from *Platycodon grandiflorum* A. DC on CRP concentration in blood serum of rats. Values are expressed as means ± SD, *—vs. control group, *p* ≤ 0.05, #—vs. control group+SO_2_, *p* ≤ 0.05

**Table 1 molecules-25-05020-t001:** Sum of saponins and inulin contents in different water extracts of *Platycodon grandiflorum* A. DC (*PG*).

Analyzed Material ^#^	Sum of Saponins [%]	Inulin [%]
E1	17.39 ± 1.30	31.19 ± 2.60
E2	24.95 ± 0.48 *	34.75 ± 0.64 *
E3	37.04 ± 0.82 *^,+^	39.68 ± 0.08 *^,+^

Legend:^#^—determined in extract dry mass, mean ± SD (*n* = 4), E1—extract from roots of field cultivated of *PG,* E2—extract from biotransformed roots of *PG*, E3—extract from callus of *PG*, *—significant difference vs. E1, *p* ≤ 0.05, +—significant difference vs. E2, *p* ≤ 0.05

**Table 2 molecules-25-05020-t002:** Secondary metabolites identified in *Platycodon grandiflorum* A. DC extracts by LC-MS systems.

No.	rt	MW	Fragmentation in Negative Ionization	Identification	Platycodon			Formula	Mass of [M-H]-		error [ppm]	PubMed	ID l	Ref
					E1	E2	E3		calculated	measured				
1	3.5	990	989, 809, 827,647, 485	caffeic acid pentaglycoside		*		C_39_H_57_O_29_	989.3214	989.32104	0.821		3	[26]
2	3.6	828	827, 665, 647, 501, 339, 179	caffeic acid tetraglycoside		*		C_33_H_47_O_24_	827.2674	827.26855	1.385		3	[26]
3	4.6	666	665, 5030,0 485,0 381,0 341, 161	caffeic acid triglycoside		*		C_24_H_41_O_21_	665.2146	665.21527	1.035		3	[26]
4	7.9	516	515, 353, 323, 191, 179	5-caffeoylquinic acid glucoside			*	C_22_H_27_O_14_	515.1406	515.13953	−2.133		2	[26]
5	8.6	354	353, 269, 250, 191, 179	3-caffeoylquinic acid		*		C_16_H_17_O_9_	353.087	353.1041	4.76		1 (std) *	[26]
6	8.7	610	609, 447, 327	isoorientin 7-*O*-glucoside	*			C_27_H_29_O_16_	609.1456	609.1477	−3.521	170474254	1 (std) *	[26]
7	9.3	610	609, 489, 429, 309	isoorientin 2”-*O*-glucoside	*			C_27_H_29_O_16_	609.145	609.1424	−4.2946		2	[26]
8	9.6	684	683, 521, 503, 491, 359, 285	lariciresinol *O*-diglucoside	*			C_32_H_43_O_16_	683.25641	683.2557	1.098		2	[27]
9	9.7	594	593, 473, 431, 311	Isovitexin 7-*O*-glucoside	*			C_27_H_29_O_15_	593.1507	593.1519	2.973	170474252	1 (std) *	[26]
10	9.8	522	567, 521, 475, 397, 341	lariciresinol *O*-glucoside	*			C_26_H_33_O_11_	521.20355	521.2036	1.372	11972395	2	[27]
11	10.4	354	353, 191, 173	5-caffeoylquinic acid			*	C_16_H_17_O_9_	353.087	353.0857	−2.8751	1794427	1 (std) *	[26]
12	10.5	588	587, 425, 263, 231, 161, 143,	n-hexyl-triglucoside		*		C_24_H_43_O_16_	587.25623	587.2557	0.966		3	[28]
13	13.4	558	557, 395, 263, 161, 143	lobetyolin glucoside		*		C_26_H_37_O_13_	557.22394	557.224	−0.044		3	[29]
14	14.3	464	463, 417, 301, 265	quercetin *O*-glucoside	*			C_21_H_19_O_12_	463.08926	463.0882	2.291		2	[26]
15	16.2	845	844, 683, 595, 409, 150	platycoside K	*			C_42_H_67_O_17_	844.31201	844.3159	−4.622	102004765	3	[30]
16	17.2	504	503, 323, 161	caffeic acid diglucoside		*		C_21_H_27_O_14_	503.14038	503.1406	−0.494		3	[28]
17	19.3	1107	1106, 991, 941, 868, 857, 749	platyconic acid C	*			C_52_H_83_O_25_	1107.52441	1107.5229	1.372	102052424	3	[31]
18	19.5	402	401, 274, 229	Icariside F	*		*	C_18_H_25_O_10_	401.14572	401.1453	0.997	C00031877	3	[28]
19	19.8	828	827, 747, 665, 501, 454	platycodigenin gentobioside	*	*	*	C_42_H_67_O_16_	827.44464	827.4435	1.427		3	[28]
20	20	960	959, 869, 798, 711, 670, 496	platycoside F	*			C_47_H_75_O_20_	959.48444	959.4857	−1.332	101048500	3	[32]
21	22.6	1400	1399, 1355, 1190, 988, 654, 572	platycoside I	*		*	C_64_H_103_O_33_	1399.63647	1399.6387	−1.599	11622299	3	[32]
22	25.9	1387	1386, 943, 681, 519, 471, 409, 376, 317,	platycodin D3		*		C_63_H_101_O_33_	1385.62158	1385.6231	−1.067	70698293	3	[32]
23	26.7	1255	1254, 843, 682, 519, 444, 375	deapi-platycodin D3	*	*		C_58_H_93_O_29_	1253.58081	1253.5808	0.008	50900942	3	[32]
24	27.1	564	563, 413, 293	apigenin 6-C-[2′′-*O*-glucoside]-arabinoside	*			C_26_H_27_O_10_	563.1395	563.1414	3.246		2	[33]
25	27.1	1428	1427,1367, 843, 825, 513	3′′-*O*-acetyl-platycodin D3		*	*	C_65_H_103_O_34_	1427.63367	1427.6336	0.033		3	[28]
26	27.7	1370	1369, 827, 665, 503, 461	polygalacin D2		*		C_63_H_101_O_32_	1369.63013	1369.6281	1.45	53325781	3	[33]
27	28.5	1296	1295, 885, 843, 643, 569	deapi-2′′-*O*-acetyl-platycodin D2		*	*	C_60_H_95_O_30_	1295.59229	1295.5914	3.41	60712775	3	[28]
28	28.6	624	623, 443, 323	isoscoparin 2′′-*O*-glucoside	*			C_28_H_31_O_16_	623.1612	623.1634	−3.5303	170474228	2	[26]
29	10.4	816	815, 461, 447, 327	isoorientin 7-*O*-[6′′-sinapoyl]-glucoside	*			C_38_H_39_O_20_	815.20422	815.204	0.254	170474256	2	[26]
30	29	1428	1427, 1368, 1277, 843, 825, 781, 663, 620, 471	3′′-*O*-acetylo-platycodin D2		*		C_65_H_103_O_34_	1427.63489	1427.6336	0.888	160712921	3	[28]
31	29	1266	1265, 1037, 877, 767, 554	platycodin C			*	C_59_H_93_O_29_	1265.57996	1265.5808	−0.663	46173919	3	[28]
32	29.2	594	593, 443, 323	isoscoparin 2′′-*O*-arabinoside	*			C_27_H_29_O_15_	593.1507	593.1519	2.973	170474209	2	[26]
33	29.6	1255	1254, 843, 663, 519, 473, 493	platycoside A		*		C_58_H_93_O_29_	1253.5813	1253.5808	0.399	50900942	3	[32]
34	29.9	1092	1091, 681, 635, 457, 407, 391, 375	deapi-platycodin D		*	*	C_52_H_83_O_24_	1091.52869	1091.528	0.654	70698266	3	[32]
35	30.5	1106	1105, 695, 519	platycoside N			*	C_53_H_84_O_24_	1104.53564	1104.5358	−0.146		3	[34]
36	31	1266	1265, 1205, 723, 681, 561, 519, 501, 471, 379	platycodin A		*	*	C_59_H_93_O_29_	1265.58459	1265.6808	2.995	46173910	3	[33]
37	31.1	1224	1223, 1133, 1014, 959, 681, 635, 633, 501, 569, 391	platycodin D		*	*	C_57_H_91_O_28_	1223.57031	1223.5702	0.061	162859	3	[33]
38	31.2	1386	1385, 843, 681, 519, 471	platycodin D2		*		C_63_H_101_O_33_	1385.62268	1385.6231	−0.273	53317652	3	[28]
39	31.3	1238	1237, 1027, 541, 485, 423, 347	deapi-polygalacin D3		*	*	C_58_H_93_O_28_	1237.58594	1237.5859	0.044		3	[28]
40	31.4	786	785, 447, 327	isoorientin 7-*O*-[6′′-feruloyl]-glucoside	*			C_37_H_37_O_19_	785.1929	785.1966	−4.687		2	[26]
41	31.5	1281	1280, 1069, 695, 521, 435, 374	platycodin K		*	*	C_59_H_92_O_30_	1279.55798	1279.5601	−1.629		3	[31]
42	31.7	1428	1427, 1367, 1206, 1021, 825, 843, 519	2′′-*O*-acetyl-deapi-platycodin D2		*		C_65_H_103_O_34_	1426.62378	1246.6238	−1.414		3	[28]
43	31.9	1267	1266, 1205, 1133, 1115, 723, 681, 663, 469	2′′-*O*-acetyl-platycodin D		*	*	C_59_H_94_O_29_	1265.58301	1265.583	1.747		3	[28]
44	32.7	844	843, 681, 519, 473, 408, 377	platycoside L		*		C_42_H_68_O_17_	843.43903	843.4384	0.778	11556931	3	[30]
45	33.3	1077	1076, 1045, 835, 791, 598, 503, 485, 427	platycoside J	*	*		C_52_H_84_O_23_	1075.53406	1075.5331	0.928	11528185	3	[31]
46	34.4	682	681, 635, 457, 519, 407	3-*O*-glucoside platycodigenin		*	*	C_36_H_57_O_12_	681.38617	681.3856	0.909		3	[28]
47	34.7	610	609, 285, 188	luteolin diglucoside			*	C_27_H_29_O_16_	609.1456	609.1477	−3.521		3	[26]
48	35.6	666	665, 619, 503	3-*O*-glucoside polygalacic acid	*	*		C_36_H_57_O_11_	665.39105	665.3906	0.623		3	[28]

Legend: a—metabolite identification level according to Metabolomics Standards Initiative recommendation [35]. The levels include: (1) Identified compounds, (2) Compounds annotated without chemical reference standards, based upon physicochemical properties and spectral similarity with public spectral libraries, (3) Compound classes characterized based upon characteristic physicochemical properties of a chemical class of compounds, or by spectral similarity to known compounds of a chemical class, (4) Unknown compounds—although unidentified or unclassified, these metabolites can still be differentiated and quantified, based upon spectral data. std—identification on the basis of standard compound fragmentation; * standard provided by Extrasynthese, France, MW—molecular mass.

**Table 3 molecules-25-05020-t003:** Influence of different extracts from *Platycodon grandiflorum* A. DC on VEGF, TGF-β1, TGF-β2 and mucin (MUC5AC) in bronchoalveolar lavage fluid (BALF) in rats.

Group	n	VEGF [pg/mL]	TGF-β1 [pg/mL]	TGF-β2 [pg/mL]	Mucin [ng/mL]
control	10	69 ± 37	175 ± 73	111 ± 5	3.92 ± 0.84
control+SO_2_	9	229 ± 53 *	404 ± 48 *	197 ± 37 *	11.5 ± 3.57 *
E1 + SO_2_	9	77 ± 38 ^#^	251 ± 49 ^#^	203 ± 11	4.04 ± 0.53 ^#^
E2 + SO_2_	9	113 ± 33^#^	192 ± 90 ^#^	195 ± 17	2.64 ± 0.15 ^#^
E3 + SO_2_	9	88 ± 56 ^#^	164 ± 25 ^#^	185 ± 55	4.31 ± 0.64 ^#^

Legend: n—number of rats. Values are expressed as means ± SD; *—vs. control group, *p* ≤ 0.05; ^#^—vs. control group+SO_2_, *p* ≤ 0.05.

**Table 4 molecules-25-05020-t004:** Influence of different extracts from *Platycodon grandiflorum* A. DC on VEGF, TGF-β1, TGF-β2, TGF-β3, and mucin (MUC5AC) mRNA expression in rats’ bronchi.

Group	n	VEGF	TGF-β1	TGF-β2	TGF-β3	Mucin
control	10	0.76 ± 0.40	1.15 ± 0.24	0.87 ± 0.35	0.73 ± 0.12	0.64 ± 0.10
Control + SO_2_	9	182 ± 52 *	1.61 ± 0.25 *	9.23 ± 0.23 *	0.61 ± 0.04	0.47 ± 0.15 *
E1 + SO_2_	9	43 ± 28 ^#^	1.06 ± 0.21 ^#^	0.75 ± 0.16 ^#^	0.54 ± 0.15	1.06 ± 0.10 ^#^
E2 + SO_2_	9	79 ± 27 ^#^	0.96 ± 0.14 ^#^	0.87 ± 0.10 ^#^	1.01 ± 0.17 ^#^	0.80 ± 0.15 ^#^
E3 + SO_2_	9	52 ± 35 ^#^	1.02 ± 0.13 ^#^	1.15 ± 0.12 ^#^	0.92 ± 0.12 ^#^	0.97 ± 0.13 ^#^

Legend: n—number of rats. Values are expressed as means±SD, *—vs. control, *p* ≤ 0.05, ^#^—vs. control+SO_2_, *p* ≤ 0.05.

**Table 5 molecules-25-05020-t005:** Influence of different extracts from *Platycodon grandiflorum* A. DC on TGF-β1, TGF-β2, TGF-β3, and mucin (MUC5AC) mRNA expression in the rats’ lungs.

Group	n	TGF-β1	TGF-β2	TGF-β3	Mucin
control	10	3.15 ± 0.75	7.48 ± 2.23	5.60 ± 1.35	0.26 ± 0.03
Control + SO_2_	9	6.39 ± 2.04 *	12.2 ± 0.81 *	1.96 ± 0.21 *	0.72 ± 0.12 *
E1 + SO_2_	9	2.34 ± 0.40 ^#^	2.03 ± 0.38 ^#^	4.72 ± 0.72 ^#^	1.14 ± 0.44 ^#^
E2 + SO_2_	9	0.56 ± 0.22^#, &^	0.45 ± 0.20 ^#, &^	1.13 ± 0.29	0.44 ± 0.09 ^#^
E3 + SO_2_	9	0.57 ± 0.13^#, &^	0.24 ± 0.10 ^#, &^	1.49 ± 0.12	1.10 ± 0.19 ^#^

Legend: n—number of rats. Values are expressed as means±SD; *—vs. control, *p* ≤ 0.05; ^#^—vs. control+SO_2_, *p* ≤ 0.05 & –vs. E1 + SO_2_, *p* ≤ 0.05

**Table 6 molecules-25-05020-t006:** Sequences of primers used for real-time PCR.

Gene	Sequence of Primers (5′–3′)	Size of the Product [pz]
GAPDH	F: GAT GGT GAA GGT CGG TGT G R: ATG AAG GGG TCG TTG ATG G	108
TGF-β1	F: CAACGCAATCTATGACAAAACC R: CTCCACAGTTGACTTGAATCT	145
TGF-β2	F: TTTGGATGCCGCCTATTGCTT R: TGAGGACTTTGGTGTGTTGTG	185
TGF-β3	F: CAAAGGAGTGGACAACGAAGA R: AGTCGGTGTGGAGGAATCAT	114
Mucin (Muc5AC)	F: TACAATGGGCAACGGTACCATCCT R: AACTGCAGGTGTCAACGATCCTCT	124
VEGF	F: GCAGACCAAAGAAAGATAGAA R: CAGTGAACGCTCCAGGATTTA	112

**Table 7 molecules-25-05020-t007:** Real-time PCR conditions for particular genes.

Gene	Number of Cycles	Initial Denaturation	Denaturation	Annealing	Elongation
GAPDH	35	95 °C, 10 min	95 °C, 4s	56 °C, 4s	72 °C, 8s
TGF-β1	40	95 °C, 10 min	95 °C, 10s	58 °C, 6s	72 °C, 10s
TGF-β2	40	95 °C, 10 min	95 °C, 10s	60 °C, 7s	72 °C, 10s
TGF-β3	40	95 °C, 10 min	95 °C, 10s	60 °C, 7s	72 °C, 10s
Mucin (Muc5AC)	40	95 °C, 10 min	95 °C, 10s	60 °C, 7s	72 °C, 10s
VEGF	40	95 °C, 10 min	95 °C, 10s	60 °C, 7s	72 °C, 10s

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
