# Peer review of "The Effect of Different Water Extracts from Platycodon grandiflorum on Selected Factors Associated with Pathogenesis of Chronic Bronchitis in Rats"

_molecules, 2020, doi:10.3390/molecules25215020_

Round 1
Reviewer 1 Report
Many grammatical errors:
Line 41-45, consider reorganizing sentences to make it easy to read. List all those elevated factors in SO2 treated rats together. Then, mention what extracts did.
Line 94-95, 131-133, 345-373 different font size?
Line 113 double space: three || extracts
Figures 1-3: chromatography conditions needs to be clearly mentioned.
Author Response
Dear Editor, Dear Reviewers,
We would like to submit a revised version of our manuscript entitled (new title – replacement of “Standardized” to “Water “as suggested by Reviewer #2):”The Effect of Different Water Extracts from Platycodon Grandiflorum on Selected Factors Associated with Pathogenesis of Chronic Bronchitis in Rats” for publication in “Molecules”. We appreciated the overall positive evaluation of our manuscript by the two reviewers and we would like to thank the reviewers for their efforts and constructive criticism that helped us to improve the quality of our manuscript. Please find below a point by point response to the issues raised by the reviewer. All changes in the manuscript are marked in yellow highlighting.
Reviewer #1
Comment 1.
Line 41-45, consider reorganizing sentences to make it easy to read. List all those elevated factors in SO2 treated rats together. Then, mention what extracts did.
Response
We agreed with the reviewer’s comment. Hence, we changed this part of the text in the abstract as suggested, and now it looks like this (lines 42-44): "In SO2-exposed rats an elevation of the CRP, TGF-β1, TGF-β2, VEGF and mucin was found, but the extracts administration mostly reversed this phenomenon, leading to control values.”
Comment 2.
Line 94-95, 131-133, 345-373 different font size?
Response
As suggested, we changed the font type and size used in the whole body text (Palatino linotype, size 10) - new lines 92-93, 128-130, 378-379, 401-413, respectively.
Comment 3.
Line 113 double space: three || extracts
Response
As suggested, we have eliminated this double space - new line 110.
Comment 4.
Figures 1-3: chromatography conditions needs to be clearly mentioned.
Response:
Currently, captions under Figures 1-3 are as follows:
“UV chromatogram of the E3 extract from Platycodon grandiflorum A. DC recorded at =230 nm (A) nm and =280 nm (B) by UPLC system. Chromatographic profiles of metabolites were obtained with gradients of two solvents: A (water acidified with 0.1% formic acid) and B (100 % acetonitrile), mobile phase flow of 0.4 ml/min on a RP C18 column at 40°C. The gradient elution: 100% of A linearly changed to 80% of A over 2 min, then to 70% of A over 8 min and to 5% of A over 1 min. The number corresponded to numbers of identified metabolites in Tab. 2”.

Reviewer 2 Report
The manuscript from Buchwald et al. evaluated the effect of tree water extracts from Platycodon grandiflorum in a model of chronic bronchitis induced by sodium metabisulfite in rats. The present manuscript is a piece of conscientious scientific work. However, the results, as authors themselves recognize are sometimes ambiguous. Therefore, some clarifications should be addressed before publishing:
- In the title, the authors are using the term standardized extracts but it seems that it has an improper use at this We have to take in account that the chemical composition of a plant strongly depends on the environment, climate, harvesting period and so on. For this reason, the specification regarding the standardized plant extracts products should include identity, the quantity of declared and identified compounds, impurities if any (e.g. degradation products) and contaminants. Therefore, in a standardized extract, the compounds considered responsible for their pharmacological action should be provided each time at the same concentration. This article uses also roots of the field-cultivated plant, and the quantities of individual compounds were not provided. Therefore I would suggest removing the term “standardized” from the title and/or replace it with another word, more appropriate.
- The use of plant name should be uniform during the manuscript - Platycodon Grandiflorum vs. Platycodi radix.
- Figure 5. Why * appears only for Contro+SO2?
- Try to avoid unclear or ambiguous formulations: line248- “other chemical substances”, line 263- “different cytokines”, line 267 – “a few cytokines”. Please be more specific
- Line 277 – typing mistake – NF-nB.
- Taking into account your affirmation from Lines 278-288 I think that you should further investigate also thinking from other perspectives. For example, what other compounds, not belonging to saponins and inulin were common to E1 and E2 extracts and might possess pharmacological effects? Also, you should try to find if you can correlate or explain the phenotypic differences with the differences in tissue expression patterns of TGFb –isoforms. The differences might be connected with the functional differences between TGF b isoforms. Can you find in vivo research articles that can support your findings?
- Please try to explain why in some cases the results were correlated with saponin or inulin concentration and in others no.
- Please use the same writing format (font size) throughout the manuscript
- Line 378-379 – you should be more specific.
- Please put the materials and methods section after the introduction
Author Response
Dear Editor, Dear Reviewers,
We would like to submit a revised version of our manuscript entitled (new title – replacement of “Standardized” to “Water “as suggested by Reviewer #2):”The Effect of Different Water Extracts from Platycodon Grandiflorum on Selected Factors Associated with Pathogenesis of Chronic Bronchitis in Rats” for publication in “Molecules”. We appreciated the overall positive evaluation of our manuscript by the two reviewers and we would like to thank the reviewers for their efforts and constructive criticism that helped us to improve the quality of our manuscript. Please find below a point by point response to the issues raised by the reviewer. All changes in the manuscript are marked in yellow highlighting.
Reviewer #2
Comment 1.
In the title, the authors are using the term standardized extracts but it seems that it has an improper use at this We have to take in account that the chemical composition of a plant strongly depends on the environment, climate, harvesting period and so on. For this reason, the specification regarding the standardized plant extracts products should include identity, the quantity of declared and identified compounds, impurities if any (e.g. degradation products) and contaminants. Therefore, in a standardized extract, the compounds considered responsible for their pharmacological action should be provided each time at the same concentration. This article uses also roots of the field-cultivated plant, and the quantities of individual compounds were not provided. Therefore I would suggest removing the term “standardized” from the title and/or replace it with another word, more appropriate.
Response
Thank you for your suggestion. Indeed, according to the definition of "Standardized", all the extracts used should be subjected a detailed phytochemical analysis with the exact specification of the compounds for which the extract is standardized (e.g. the most characteristic, most abundant or having a specific pharmacological effect) and any other information that accurately describes the preparation of plant origin. It seems, however, that at this stage of knowledge of the composition of these extracts, such a procedure is impossible, without knowing all the properties of the tested extracts, which was partly the aim of the presented research. Hence, agreeing with the reviewer's remark, we decided to change this term to "Water", thus extending the title of the information on the type of extracts being the subject of the presented research.
Comment 2.
The use of plant name should be uniform during the manuscript - Platycodon Grandiflorum vs. Platycodi radix.
Response
As suggested, we unified the nomenclature throughout the text of the work by replacing "Platycodi radix" by using „ root of Platycodon glandiflorum” (for example: new lines 65, 276, 277, 286).
Comment 3.
Figure 5. Why * appears only for Contro+SO2?
Response
The * sign was used to define the occurrence of the difference for the value that was statistically significant in relation to the control group (it means rats that were not exposed to MBS = appropriate control negative study group) at the significance level p≤0.05. In fact, this * sign was only used to denote the difference to the value for positive control group (it means animals exposed to SO2 (= control + SO2)), thus emphasizing the effect of SO2. At the same time, other groups, i.e. those treated with SO2 and extract, were not marked with this sign intentionally. From the methodological point of view, it seems that in this way it is avoided to emphasize the effect of two factors at the same time, i.e. SO2 and the extract in comparison to the control negative group (control), because when analyzing such a phenomenon it is not possible to say clearly which of these two factors is important in relation to the control negative group (control). This method of marking significance was also used for the remaining results, i.e. in the appropriate tables.
Comment 4.
Try to avoid unclear or ambiguous formulations: line248- “other chemical substances”, line 263- “different cytokines”, line 267 – “a few cytokines”. Please be more specific.
Response
Thank you for your suggestion. Following your advice, we have revised these terms as follows:
- line 248 (new line 261) by deleting "other chemical substances" and now this sentence is changed as follows) ”Inflammatory reactions in the bronchial mucosa in respiratory tract diseases are controlled by set of interacting factors, e.g. cytokines.”
- line 263 (new line 274) by deleting “different” and now this sentence is changed as follows: “It is in accordance with publications by many authors, which clearly demonstrated that animals exposed to SO2 develop chronic inflammatory processes which are sustained by pro-inflammatory mediators, mainly cytokines”.
- line 267 (new line 278) by inserting “ Th1 and Th2 cytokines“ and now this sentence is changed as follows: “Extracts from the roots of PG markedly decreased the number of infiltrated inflammatory cells and the levels of Th1 and Th2 cytokines and chemokines compared with a control group and reduced ovalbumin-specific IgE levels in BALF fluid [9]”.
Comment 5.
Line 277 – typing mistake – NF-nB.
Response
Thank you for your suggestion. It was our typographic mistake. We changed this error to the correct wording of this factor in "NF-kB" (new line 288).
Comment 6.
Taking into account your affirmation from Lines 278-288 I think that you should further investigate also thinking from other perspectives. For example, what other compounds, not belonging to saponins and inulin were common to E1 and E2 extracts and might possess pharmacological effects? Also, you should try to find if you can correlate or explain the phenotypic differences with the differences in tissue expression patterns of TGFb –isoforms. The differences might be connected with the functional differences between TGF b isoforms. Can you find in vivo research articles that can support your findings?
Response
Thank you for your suggestion.
In trying to answer your questions, we have introduced new paragraphs in the text which we hope sheds some light on the issues raised in these.
- The answer to the question:„For example, what other compounds, not belonging to saponins and inulin were common to E1 and E2 extracts and might possess pharmacological effects?” is as follows:
New lines 299-314:
“The differences obtained in this study are not only related to different content of saponins and inulin (Tab. 1), but also to a slightly different qualitative composition of the examined extracts, as mentioned in the Results section (chapter 2.2. Qualitative analysis). This may be in line with the observation, that only one compound, No. 19, (platycodigenin gentobioside) is common to all extracts (Fig. 4). In addition to saponins (No. 15, 17, 19, 20, 21, 22, 23, 25, 26, 27, 30, 31, 33, 34, 35, 36, 37, 38, 39, 41, 42, 43, 44, 45, 46, 48), the presence of other groups of chemical compounds can be noticed (flavonoids: 6, 7, 9, 14, 24, 28, 29, 32, 40, 47; hydroxycinnamic acids: 1, 2, 3, 4, 5, 11, 16; lignans: 8, 10, saccharide: 12), which varied significantly among samples from the extracts (for example: flavonoids only in E1 extract). This is consistent with the knowledge of the possible contribution of these compounds present in PG to various pharmacological effects on inflammatory basis [58]. Hence, it cannot be ruled out that the observed differences in anti-inflammatory activity may have been caused by the presence of flavonoid glycosides, which were identified mainly in the E1 extract, whereas caffeic acid glycosides and 3-caffeoylquinic acid were characteristic only for E2 extract, and the isomer 5-caffeoylquinic acid and its glycoside were detected in the E3 extract (Tab. 3). Nevertheless, their possible significance should be supported by detailed quantification (content) of the above-mentioned compounds in the future”.
- The answer to the question:” Also, you should try to find if you can correlate or explain the phenotypic differences with the differences in tissue expression patterns of TGFb –isoforms. The differences might be connected with the functional differences between TGF b isoforms. Can you find in vivo research articles that can support your findings?” is as follows:
New lines 318-320:”The isoform TGF-β1 is responsible for fibrinogen activity in chronic bronchitis and plays a role in airway wall remodeling [2]. Moreover, this isoform produced the inhibitory action on immune cell differentiation (Th1 and Th2 cells and B cells) and cytokine production (IFN-? and IL-2) [59]”.
New lines 343-362:”The differences observed in the response of TGF-β isoforms to the action of SO2 and the administration of extracts may be related to a slightly different distribution and action of those in airways, since TGF-β1 colocalizes with extracellular matrix proteins, such as collagen, and interfaces between epithelial and mesenchymal cells, TGF-β2 is found in endodermal bronchiolar epithelium, whereas TGF-β3 is expressed in tracheal mesenchyme and the endodermal epithelial cells in bronchioles and mesodermal cells [60]. Moreover, the biological effects of different TGF-β isoforms depend on their availability, combination of two types of their receptors, and intracellular signaling pathways that they can induce [54]. For example, in the airways of humans with asthma, TGF-β1 levels are elevated as compared with normal control subjects, suggesting a role in the repair of injured asthmatic airways or the existence of a negative feedback loop controlling airway inflammation [60]. Similarly, an increase in TGF-β2 expression in asthmatic epithelium was shown, which correlates with an increase in the number of eosinophils and neutrophils in patients with severe and mild asthma [54]. There is relatively little information on TGF-β3 expression, although the available data suggests that there is no difference in TGF-β3 expression between asthmatic patients and control subjects [54], but in patients with severe COPD the lowering of TGF-β3 was found [55]. In conclusion, SO2, as a pro-inflammatory factor, induced changes in the expression of individual TGF-β isoforms in a similar way as the above-mentioned. However, the result of administration of the extracts to individual isoforms is not easily explained and will likely, as it was mentioned previously, requires in-depth research on the effect of not only saponins and inulin occurring in various amounts in the studied extracts, but also compounds from other chemical groups”.
- We found some information in additional cited works [new items -59 and 60].
Comment 7.
Please try to explain why in some cases the results were correlated with saponin or inulin concentration and in others no.
Response
We think that we have responded to this comment in some way in the first part of our reply to Comment 6.
Comment 8.
Please use the same writing format (font size) throughout the manuscript
Response
This note is of the same type as Reviewer # 1, Comment 2 and as suggested, we changed the font type and size used in the whole body text (Palatino linotype, size 10) - new lines 92-93, 128-130, 378-379, 401-413, respectively.
Comment 9.
Line 378-379 – you should be more specific.
Response
Thank you for your suggestion. However, due to the fact that this part of the text (new lines 417-418) is a kind of conclusion, it seems that such a generalizing nature of the sentence is justified at this point.
Comment 10.
Please put the materials and methods section after the introduction.
Response
Order of the chapters of this work was presented in accordance with the guidelines on the website of the "Molecules" journal (https://www.mdpi.com/journal/molecules/instructions#preparation - Instructions for Authors - Research Manuscript Sections) that is: Introduction, Results, Discussion, Materials and Methods, Conclusions.
Dear Editor, Dear Reviewers,
We would like to submit a revised version of our manuscript entitled (new title – replacement of “Standardized” to “Water “as suggested by Reviewer #2):”The Effect of Different Water Extracts from Platycodon Grandiflorum on Selected Factors Associated with Pathogenesis of Chronic Bronchitis in Rats” for publication in “Molecules”. We appreciated the overall positive evaluation of our manuscript by the two reviewers and we would like to thank the reviewers for their efforts and constructive criticism that helped us to improve the quality of our manuscript. Please find below a point by point response to the issues raised by the reviewer. All changes in the manuscript are marked in yellow highlighting.
Reviewer #2
Comment 1.
In the title, the authors are using the term standardized extracts but it seems that it has an improper use at this We have to take in account that the chemical composition of a plant strongly depends on the environment, climate, harvesting period and so on. For this reason, the specification regarding the standardized plant extracts products should include identity, the quantity of declared and identified compounds, impurities if any (e.g. degradation products) and contaminants. Therefore, in a standardized extract, the compounds considered responsible for their pharmacological action should be provided each time at the same concentration. This article uses also roots of the field-cultivated plant, and the quantities of individual compounds were not provided. Therefore I would suggest removing the term “standardized” from the title and/or replace it with another word, more appropriate.
Response
Thank you for your suggestion. Indeed, according to the definition of "Standardized", all the extracts used should be subjected a detailed phytochemical analysis with the exact specification of the compounds for which the extract is standardized (e.g. the most characteristic, most abundant or having a specific pharmacological effect) and any other information that accurately describes the preparation of plant origin. It seems, however, that at this stage of knowledge of the composition of these extracts, such a procedure is impossible, without knowing all the properties of the tested extracts, which was partly the aim of the presented research. Hence, agreeing with the reviewer's remark, we decided to change this term to "Water", thus extending the title of the information on the type of extracts being the subject of the presented research.
Comment 2.
The use of plant name should be uniform during the manuscript - Platycodon Grandiflorum vs. Platycodi radix.
Response
As suggested, we unified the nomenclature throughout the text of the work by replacing "Platycodi radix" by using „ root of Platycodon glandiflorum” (for example: new lines 65, 276, 277, 286).
Comment 3.
Figure 5. Why * appears only for Contro+SO2?
Response
The * sign was used to define the occurrence of the difference for the value that was statistically significant in relation to the control group (it means rats that were not exposed to MBS = appropriate control negative study group) at the significance level p≤0.05. In fact, this * sign was only used to denote the difference to the value for positive control group (it means animals exposed to SO2 (= control + SO2)), thus emphasizing the effect of SO2. At the same time, other groups, i.e. those treated with SO2 and extract, were not marked with this sign intentionally. From the methodological point of view, it seems that in this way it is avoided to emphasize the effect of two factors at the same time, i.e. SO2 and the extract in comparison to the control negative group (control), because when analyzing such a phenomenon it is not possible to say clearly which of these two factors is important in relation to the control negative group (control). This method of marking significance was also used for the remaining results, i.e. in the appropriate tables.
Comment 4.
Try to avoid unclear or ambiguous formulations: line248- “other chemical substances”, line 263- “different cytokines”, line 267 – “a few cytokines”. Please be more specific.
Response
Thank you for your suggestion. Following your advice, we have revised these terms as follows:
- line 248 (new line 261) by deleting "other chemical substances" and now this sentence is changed as follows) ”Inflammatory reactions in the bronchial mucosa in respiratory tract diseases are controlled by set of interacting factors, e.g. cytokines.”
- line 263 (new line 274) by deleting “different” and now this sentence is changed as follows: “It is in accordance with publications by many authors, which clearly demonstrated that animals exposed to SO2 develop chronic inflammatory processes which are sustained by pro-inflammatory mediators, mainly cytokines”.
- line 267 (new line 278) by inserting “ Th1 and Th2 cytokines“ and now this sentence is changed as follows: “Extracts from the roots of PG markedly decreased the number of infiltrated inflammatory cells and the levels of Th1 and Th2 cytokines and chemokines compared with a control group and reduced ovalbumin-specific IgE levels in BALF fluid [9]”.
Comment 5.
Line 277 – typing mistake – NF-nB.
Response
Thank you for your suggestion. It was our typographic mistake. We changed this error to the correct wording of this factor in "NF-kB" (new line 288).
Comment 6.
Taking into account your affirmation from Lines 278-288 I think that you should further investigate also thinking from other perspectives. For example, what other compounds, not belonging to saponins and inulin were common to E1 and E2 extracts and might possess pharmacological effects? Also, you should try to find if you can correlate or explain the phenotypic differences with the differences in tissue expression patterns of TGFb –isoforms. The differences might be connected with the functional differences between TGF b isoforms. Can you find in vivo research articles that can support your findings?
Response
Thank you for your suggestion.
In trying to answer your questions, we have introduced new paragraphs in the text which we hope sheds some light on the issues raised in these.
- The answer to the question:„For example, what other compounds, not belonging to saponins and inulin were common to E1 and E2 extracts and might possess pharmacological effects?” is as follows:
New lines 299-314:
“The differences obtained in this study are not only related to different content of saponins and inulin (Tab. 1), but also to a slightly different qualitative composition of the examined extracts, as mentioned in the Results section (chapter 2.2. Qualitative analysis). This may be in line with the observation, that only one compound, No. 19, (platycodigenin gentobioside) is common to all extracts (Fig. 4). In addition to saponins (No. 15, 17, 19, 20, 21, 22, 23, 25, 26, 27, 30, 31, 33, 34, 35, 36, 37, 38, 39, 41, 42, 43, 44, 45, 46, 48), the presence of other groups of chemical compounds can be noticed (flavonoids: 6, 7, 9, 14, 24, 28, 29, 32, 40, 47; hydroxycinnamic acids: 1, 2, 3, 4, 5, 11, 16; lignans: 8, 10, saccharide: 12), which varied significantly among samples from the extracts (for example: flavonoids only in E1 extract). This is consistent with the knowledge of the possible contribution of these compounds present in PG to various pharmacological effects on inflammatory basis [58]. Hence, it cannot be ruled out that the observed differences in anti-inflammatory activity may have been caused by the presence of flavonoid glycosides, which were identified mainly in the E1 extract, whereas caffeic acid glycosides and 3-caffeoylquinic acid were characteristic only for E2 extract, and the isomer 5-caffeoylquinic acid and its glycoside were detected in the E3 extract (Tab. 3). Nevertheless, their possible significance should be supported by detailed quantification (content) of the above-mentioned compounds in the future”.
- The answer to the question:” Also, you should try to find if you can correlate or explain the phenotypic differences with the differences in tissue expression patterns of TGFb –isoforms. The differences might be connected with the functional differences between TGF b isoforms. Can you find in vivo research articles that can support your findings?” is as follows:
New lines 318-320:”The isoform TGF-β1 is responsible for fibrinogen activity in chronic bronchitis and plays a role in airway wall remodeling [2]. Moreover, this isoform produced the inhibitory action on immune cell differentiation (Th1 and Th2 cells and B cells) and cytokine production (IFN-? and IL-2) [59]”.
New lines 343-362:”The differences observed in the response of TGF-β isoforms to the action of SO2 and the administration of extracts may be related to a slightly different distribution and action of those in airways, since TGF-β1 colocalizes with extracellular matrix proteins, such as collagen, and interfaces between epithelial and mesenchymal cells, TGF-β2 is found in endodermal bronchiolar epithelium, whereas TGF-β3 is expressed in tracheal mesenchyme and the endodermal epithelial cells in bronchioles and mesodermal cells [60]. Moreover, the biological effects of different TGF-β isoforms depend on their availability, combination of two types of their receptors, and intracellular signaling pathways that they can induce [54]. For example, in the airways of humans with asthma, TGF-β1 levels are elevated as compared with normal control subjects, suggesting a role in the repair of injured asthmatic airways or the existence of a negative feedback loop controlling airway inflammation [60]. Similarly, an increase in TGF-β2 expression in asthmatic epithelium was shown, which correlates with an increase in the number of eosinophils and neutrophils in patients with severe and mild asthma [54]. There is relatively little information on TGF-β3 expression, although the available data suggests that there is no difference in TGF-β3 expression between asthmatic patients and control subjects [54], but in patients with severe COPD the lowering of TGF-β3 was found [55]. In conclusion, SO2, as a pro-inflammatory factor, induced changes in the expression of individual TGF-β isoforms in a similar way as the above-mentioned. However, the result of administration of the extracts to individual isoforms is not easily explained and will likely, as it was mentioned previously, requires in-depth research on the effect of not only saponins and inulin occurring in various amounts in the studied extracts, but also compounds from other chemical groups”.
- We found some information in additional cited works [new items -59 and 60].
Comment 7.
Please try to explain why in some cases the results were correlated with saponin or inulin concentration and in others no.
Response
We think that we have responded to this comment in some way in the first part of our reply to Comment 6.
Comment 8.
Please use the same writing format (font size) throughout the manuscript
Response
This note is of the same type as Reviewer # 1, Comment 2 and as suggested, we changed the font type and size used in the whole body text (Palatino linotype, size 10) - new lines 92-93, 128-130, 378-379, 401-413, respectively.
Comment 9.
Line 378-379 – you should be more specific.
Response
Thank you for your suggestion. However, due to the fact that this part of the text (new lines 417-418) is a kind of conclusion, it seems that such a generalizing nature of the sentence is justified at this point.
Comment 10.
Please put the materials and methods section after the introduction.
Response
Order of the chapters of this work was presented in accordance with the guidelines on the website of the "Molecules" journal (https://www.mdpi.com/journal/molecules/instructions#preparation - Instructions for Authors - Research Manuscript Sections) that is: Introduction, Results, Discussion, Materials and Methods, Conclusions.
Dear Editor, Dear Reviewers,
We would like to submit a revised version of our manuscript entitled (new title – replacement of “Standardized” to “Water “as suggested by Reviewer #2):”The Effect of Different Water Extracts from Platycodon Grandiflorum on Selected Factors Associated with Pathogenesis of Chronic Bronchitis in Rats” for publication in “Molecules”. We appreciated the overall positive evaluation of our manuscript by the two reviewers and we would like to thank the reviewers for their efforts and constructive criticism that helped us to improve the quality of our manuscript. Please find below a point by point response to the issues raised by the reviewer. All changes in the manuscript are marked in yellow highlighting.
Reviewer #2
Comment 1.
In the title, the authors are using the term standardized extracts but it seems that it has an improper use at this We have to take in account that the chemical composition of a plant strongly depends on the environment, climate, harvesting period and so on. For this reason, the specification regarding the standardized plant extracts products should include identity, the quantity of declared and identified compounds, impurities if any (e.g. degradation products) and contaminants. Therefore, in a standardized extract, the compounds considered responsible for their pharmacological action should be provided each time at the same concentration. This article uses also roots of the field-cultivated plant, and the quantities of individual compounds were not provided. Therefore I would suggest removing the term “standardized” from the title and/or replace it with another word, more appropriate.
Response
Thank you for your suggestion. Indeed, according to the definition of "Standardized", all the extracts used should be subjected a detailed phytochemical analysis with the exact specification of the compounds for which the extract is standardized (e.g. the most characteristic, most abundant or having a specific pharmacological effect) and any other information that accurately describes the preparation of plant origin. It seems, however, that at this stage of knowledge of the composition of these extracts, such a procedure is impossible, without knowing all the properties of the tested extracts, which was partly the aim of the presented research. Hence, agreeing with the reviewer's remark, we decided to change this term to "Water", thus extending the title of the information on the type of extracts being the subject of the presented research.
Comment 2.
The use of plant name should be uniform during the manuscript - Platycodon Grandiflorum vs. Platycodi radix.
Response
As suggested, we unified the nomenclature throughout the text of the work by replacing "Platycodi radix" by using „ root of Platycodon glandiflorum” (for example: new lines 65, 276, 277, 286).
Comment 3.
Figure 5. Why * appears only for Contro+SO2?
Response
The * sign was used to define the occurrence of the difference for the value that was statistically significant in relation to the control group (it means rats that were not exposed to MBS = appropriate control negative study group) at the significance level p≤0.05. In fact, this * sign was only used to denote the difference to the value for positive control group (it means animals exposed to SO2 (= control + SO2)), thus emphasizing the effect of SO2. At the same time, other groups, i.e. those treated with SO2 and extract, were not marked with this sign intentionally. From the methodological point of view, it seems that in this way it is avoided to emphasize the effect of two factors at the same time, i.e. SO2 and the extract in comparison to the control negative group (control), because when analyzing such a phenomenon it is not possible to say clearly which of these two factors is important in relation to the control negative group (control). This method of marking significance was also used for the remaining results, i.e. in the appropriate tables.
Comment 4.
Try to avoid unclear or ambiguous formulations: line248- “other chemical substances”, line 263- “different cytokines”, line 267 – “a few cytokines”. Please be more specific.
Response
Thank you for your suggestion. Following your advice, we have revised these terms as follows:
- line 248 (new line 261) by deleting "other chemical substances" and now this sentence is changed as follows) ”Inflammatory reactions in the bronchial mucosa in respiratory tract diseases are controlled by set of interacting factors, e.g. cytokines.”
- line 263 (new line 274) by deleting “different” and now this sentence is changed as follows: “It is in accordance with publications by many authors, which clearly demonstrated that animals exposed to SO2 develop chronic inflammatory processes which are sustained by pro-inflammatory mediators, mainly cytokines”.
- line 267 (new line 278) by inserting “ Th1 and Th2 cytokines“ and now this sentence is changed as follows: “Extracts from the roots of PG markedly decreased the number of infiltrated inflammatory cells and the levels of Th1 and Th2 cytokines and chemokines compared with a control group and reduced ovalbumin-specific IgE levels in BALF fluid [9]”.
Comment 5.
Line 277 – typing mistake – NF-nB.
Response
Thank you for your suggestion. It was our typographic mistake. We changed this error to the correct wording of this factor in "NF-kB" (new line 288).
Comment 6.
Taking into account your affirmation from Lines 278-288 I think that you should further investigate also thinking from other perspectives. For example, what other compounds, not belonging to saponins and inulin were common to E1 and E2 extracts and might possess pharmacological effects? Also, you should try to find if you can correlate or explain the phenotypic differences with the differences in tissue expression patterns of TGFb –isoforms. The differences might be connected with the functional differences between TGF b isoforms. Can you find in vivo research articles that can support your findings?
Response
Thank you for your suggestion.
In trying to answer your questions, we have introduced new paragraphs in the text which we hope sheds some light on the issues raised in these.
- The answer to the question:„For example, what other compounds, not belonging to saponins and inulin were common to E1 and E2 extracts and might possess pharmacological effects?” is as follows:
New lines 299-314:
“The differences obtained in this study are not only related to different content of saponins and inulin (Tab. 1), but also to a slightly different qualitative composition of the examined extracts, as mentioned in the Results section (chapter 2.2. Qualitative analysis). This may be in line with the observation, that only one compound, No. 19, (platycodigenin gentobioside) is common to all extracts (Fig. 4). In addition to saponins (No. 15, 17, 19, 20, 21, 22, 23, 25, 26, 27, 30, 31, 33, 34, 35, 36, 37, 38, 39, 41, 42, 43, 44, 45, 46, 48), the presence of other groups of chemical compounds can be noticed (flavonoids: 6, 7, 9, 14, 24, 28, 29, 32, 40, 47; hydroxycinnamic acids: 1, 2, 3, 4, 5, 11, 16; lignans: 8, 10, saccharide: 12), which varied significantly among samples from the extracts (for example: flavonoids only in E1 extract). This is consistent with the knowledge of the possible contribution of these compounds present in PG to various pharmacological effects on inflammatory basis [58]. Hence, it cannot be ruled out that the observed differences in anti-inflammatory activity may have been caused by the presence of flavonoid glycosides, which were identified mainly in the E1 extract, whereas caffeic acid glycosides and 3-caffeoylquinic acid were characteristic only for E2 extract, and the isomer 5-caffeoylquinic acid and its glycoside were detected in the E3 extract (Tab. 3). Nevertheless, their possible significance should be supported by detailed quantification (content) of the above-mentioned compounds in the future”.
- The answer to the question:” Also, you should try to find if you can correlate or explain the phenotypic differences with the differences in tissue expression patterns of TGFb –isoforms. The differences might be connected with the functional differences between TGF b isoforms. Can you find in vivo research articles that can support your findings?” is as follows:
New lines 318-320:”The isoform TGF-β1 is responsible for fibrinogen activity in chronic bronchitis and plays a role in airway wall remodeling [2]. Moreover, this isoform produced the inhibitory action on immune cell differentiation (Th1 and Th2 cells and B cells) and cytokine production (IFN-? and IL-2) [59]”.
New lines 343-362:”The differences observed in the response of TGF-β isoforms to the action of SO2 and the administration of extracts may be related to a slightly different distribution and action of those in airways, since TGF-β1 colocalizes with extracellular matrix proteins, such as collagen, and interfaces between epithelial and mesenchymal cells, TGF-β2 is found in endodermal bronchiolar epithelium, whereas TGF-β3 is expressed in tracheal mesenchyme and the endodermal epithelial cells in bronchioles and mesodermal cells [60]. Moreover, the biological effects of different TGF-β isoforms depend on their availability, combination of two types of their receptors, and intracellular signaling pathways that they can induce [54]. For example, in the airways of humans with asthma, TGF-β1 levels are elevated as compared with normal control subjects, suggesting a role in the repair of injured asthmatic airways or the existence of a negative feedback loop controlling airway inflammation [60]. Similarly, an increase in TGF-β2 expression in asthmatic epithelium was shown, which correlates with an increase in the number of eosinophils and neutrophils in patients with severe and mild asthma [54]. There is relatively little information on TGF-β3 expression, although the available data suggests that there is no difference in TGF-β3 expression between asthmatic patients and control subjects [54], but in patients with severe COPD the lowering of TGF-β3 was found [55]. In conclusion, SO2, as a pro-inflammatory factor, induced changes in the expression of individual TGF-β isoforms in a similar way as the above-mentioned. However, the result of administration of the extracts to individual isoforms is not easily explained and will likely, as it was mentioned previously, requires in-depth research on the effect of not only saponins and inulin occurring in various amounts in the studied extracts, but also compounds from other chemical groups”.
- We found some information in additional cited works [new items -59 and 60].
Comment 7.
Please try to explain why in some cases the results were correlated with saponin or inulin concentration and in others no.
Response
We think that we have responded to this comment in some way in the first part of our reply to Comment 6.
Comment 8.
Please use the same writing format (font size) throughout the manuscript
Response
This note is of the same type as Reviewer # 1, Comment 2 and as suggested, we changed the font type and size used in the whole body text (Palatino linotype, size 10) - new lines 92-93, 128-130, 378-379, 401-413, respectively.
Comment 9.
Line 378-379 – you should be more specific.
Response
Thank you for your suggestion. However, due to the fact that this part of the text (new lines 417-418) is a kind of conclusion, it seems that such a generalizing nature of the sentence is justified at this point.
Comment 10.
Please put the materials and methods section after the introduction.
Response
Order of the chapters of this work was presented in accordance with the guidelines on the website of the "Molecules" journal (https://www.mdpi.com/journal/molecules/instructions#preparation - Instructions for Authors - Research Manuscript Sections) that is: Introduction, Results, Discussion, Materials and Methods, Conclusions.
Dear Editor, Dear Reviewers,
We would like to submit a revised version of our manuscript entitled (new title – replacement of “Standardized” to “Water “as suggested by Reviewer #2):”The Effect of Different Water Extracts from Platycodon Grandiflorum on Selected Factors Associated with Pathogenesis of Chronic Bronchitis in Rats” for publication in “Molecules”. We appreciated the overall positive evaluation of our manuscript by the two reviewers and we would like to thank the reviewers for their efforts and constructive criticism that helped us to improve the quality of our manuscript. Please find below a point by point response to the issues raised by the reviewer. All changes in the manuscript are marked in yellow highlighting.
Reviewer #2
Comment 1.
In the title, the authors are using the term standardized extracts but it seems that it has an improper use at this We have to take in account that the chemical composition of a plant strongly depends on the environment, climate, harvesting period and so on. For this reason, the specification regarding the standardized plant extracts products should include identity, the quantity of declared and identified compounds, impurities if any (e.g. degradation products) and contaminants. Therefore, in a standardized extract, the compounds considered responsible for their pharmacological action should be provided each time at the same concentration. This article uses also roots of the field-cultivated plant, and the quantities of individual compounds were not provided. Therefore I would suggest removing the term “standardized” from the title and/or replace it with another word, more appropriate.
Response
Thank you for your suggestion. Indeed, according to the definition of "Standardized", all the extracts used should be subjected a detailed phytochemical analysis with the exact specification of the compounds for which the extract is standardized (e.g. the most characteristic, most abundant or having a specific pharmacological effect) and any other information that accurately describes the preparation of plant origin. It seems, however, that at this stage of knowledge of the composition of these extracts, such a procedure is impossible, without knowing all the properties of the tested extracts, which was partly the aim of the presented research. Hence, agreeing with the reviewer's remark, we decided to change this term to "Water", thus extending the title of the information on the type of extracts being the subject of the presented research.
Comment 2.
The use of plant name should be uniform during the manuscript - Platycodon Grandiflorum vs. Platycodi radix.
Response
As suggested, we unified the nomenclature throughout the text of the work by replacing "Platycodi radix" by using „ root of Platycodon glandiflorum” (for example: new lines 65, 276, 277, 286).
Comment 3.
Figure 5. Why * appears only for Contro+SO2?
Response
The * sign was used to define the occurrence of the difference for the value that was statistically significant in relation to the control group (it means rats that were not exposed to MBS = appropriate control negative study group) at the significance level p≤0.05. In fact, this * sign was only used to denote the difference to the value for positive control group (it means animals exposed to SO2 (= control + SO2)), thus emphasizing the effect of SO2. At the same time, other groups, i.e. those treated with SO2 and extract, were not marked with this sign intentionally. From the methodological point of view, it seems that in this way it is avoided to emphasize the effect of two factors at the same time, i.e. SO2 and the extract in comparison to the control negative group (control), because when analyzing such a phenomenon it is not possible to say clearly which of these two factors is important in relation to the control negative group (control). This method of marking significance was also used for the remaining results, i.e. in the appropriate tables.
Comment 4.
Try to avoid unclear or ambiguous formulations: line248- “other chemical substances”, line 263- “different cytokines”, line 267 – “a few cytokines”. Please be more specific.
Response
Thank you for your suggestion. Following your advice, we have revised these terms as follows:
- line 248 (new line 261) by deleting "other chemical substances" and now this sentence is changed as follows) ”Inflammatory reactions in the bronchial mucosa in respiratory tract diseases are controlled by set of interacting factors, e.g. cytokines.”
- line 263 (new line 274) by deleting “different” and now this sentence is changed as follows: “It is in accordance with publications by many authors, which clearly demonstrated that animals exposed to SO2 develop chronic inflammatory processes which are sustained by pro-inflammatory mediators, mainly cytokines”.
- line 267 (new line 278) by inserting “ Th1 and Th2 cytokines“ and now this sentence is changed as follows: “Extracts from the roots of PG markedly decreased the number of infiltrated inflammatory cells and the levels of Th1 and Th2 cytokines and chemokines compared with a control group and reduced ovalbumin-specific IgE levels in BALF fluid [9]”.
Comment 5.
Line 277 – typing mistake – NF-nB.
Response
Thank you for your suggestion. It was our typographic mistake. We changed this error to the correct wording of this factor in "NF-kB" (new line 288).
Comment 6.
Taking into account your affirmation from Lines 278-288 I think that you should further investigate also thinking from other perspectives. For example, what other compounds, not belonging to saponins and inulin were common to E1 and E2 extracts and might possess pharmacological effects? Also, you should try to find if you can correlate or explain the phenotypic differences with the differences in tissue expression patterns of TGFb –isoforms. The differences might be connected with the functional differences between TGF b isoforms. Can you find in vivo research articles that can support your findings?
Response
Thank you for your suggestion.
In trying to answer your questions, we have introduced new paragraphs in the text which we hope sheds some light on the issues raised in these.
- The answer to the question:„For example, what other compounds, not belonging to saponins and inulin were common to E1 and E2 extracts and might possess pharmacological effects?” is as follows:
New lines 299-314:
“The differences obtained in this study are not only related to different content of saponins and inulin (Tab. 1), but also to a slightly different qualitative composition of the examined extracts, as mentioned in the Results section (chapter 2.2. Qualitative analysis). This may be in line with the observation, that only one compound, No. 19, (platycodigenin gentobioside) is common to all extracts (Fig. 4). In addition to saponins (No. 15, 17, 19, 20, 21, 22, 23, 25, 26, 27, 30, 31, 33, 34, 35, 36, 37, 38, 39, 41, 42, 43, 44, 45, 46, 48), the presence of other groups of chemical compounds can be noticed (flavonoids: 6, 7, 9, 14, 24, 28, 29, 32, 40, 47; hydroxycinnamic acids: 1, 2, 3, 4, 5, 11, 16; lignans: 8, 10, saccharide: 12), which varied significantly among samples from the extracts (for example: flavonoids only in E1 extract). This is consistent with the knowledge of the possible contribution of these compounds present in PG to various pharmacological effects on inflammatory basis [58]. Hence, it cannot be ruled out that the observed differences in anti-inflammatory activity may have been caused by the presence of flavonoid glycosides, which were identified mainly in the E1 extract, whereas caffeic acid glycosides and 3-caffeoylquinic acid were characteristic only for E2 extract, and the isomer 5-caffeoylquinic acid and its glycoside were detected in the E3 extract (Tab. 3). Nevertheless, their possible significance should be supported by detailed quantification (content) of the above-mentioned compounds in the future”.
- The answer to the question:” Also, you should try to find if you can correlate or explain the phenotypic differences with the differences in tissue expression patterns of TGFb –isoforms. The differences might be connected with the functional differences between TGF b isoforms. Can you find in vivo research articles that can support your findings?” is as follows:
New lines 318-320:”The isoform TGF-β1 is responsible for fibrinogen activity in chronic bronchitis and plays a role in airway wall remodeling [2]. Moreover, this isoform produced the inhibitory action on immune cell differentiation (Th1 and Th2 cells and B cells) and cytokine production (IFN-? and IL-2) [59]”.
New lines 343-362:”The differences observed in the response of TGF-β isoforms to the action of SO2 and the administration of extracts may be related to a slightly different distribution and action of those in airways, since TGF-β1 colocalizes with extracellular matrix proteins, such as collagen, and interfaces between epithelial and mesenchymal cells, TGF-β2 is found in endodermal bronchiolar epithelium, whereas TGF-β3 is expressed in tracheal mesenchyme and the endodermal epithelial cells in bronchioles and mesodermal cells [60]. Moreover, the biological effects of different TGF-β isoforms depend on their availability, combination of two types of their receptors, and intracellular signaling pathways that they can induce [54]. For example, in the airways of humans with asthma, TGF-β1 levels are elevated as compared with normal control subjects, suggesting a role in the repair of injured asthmatic airways or the existence of a negative feedback loop controlling airway inflammation [60]. Similarly, an increase in TGF-β2 expression in asthmatic epithelium was shown, which correlates with an increase in the number of eosinophils and neutrophils in patients with severe and mild asthma [54]. There is relatively little information on TGF-β3 expression, although the available data suggests that there is no difference in TGF-β3 expression between asthmatic patients and control subjects [54], but in patients with severe COPD the lowering of TGF-β3 was found [55]. In conclusion, SO2, as a pro-inflammatory factor, induced changes in the expression of individual TGF-β isoforms in a similar way as the above-mentioned. However, the result of administration of the extracts to individual isoforms is not easily explained and will likely, as it was mentioned previously, requires in-depth research on the effect of not only saponins and inulin occurring in various amounts in the studied extracts, but also compounds from other chemical groups”.
- We found some information in additional cited works [new items -59 and 60].
Comment 7.
Please try to explain why in some cases the results were correlated with saponin or inulin concentration and in others no.
Response
We think that we have responded to this comment in some way in the first part of our reply to Comment 6.
Comment 8.
Please use the same writing format (font size) throughout the manuscript
Response
This note is of the same type as Reviewer # 1, Comment 2 and as suggested, we changed the font type and size used in the whole body text (Palatino linotype, size 10) - new lines 92-93, 128-130, 378-379, 401-413, respectively.
Comment 9.
Line 378-379 – you should be more specific.
Response
Thank you for your suggestion. However, due to the fact that this part of the text (new lines 417-418) is a kind of conclusion, it seems that such a generalizing nature of the sentence is justified at this point.
Comment 10.
Please put the materials and methods section after the introduction.
Response
Order of the chapters of this work was presented in accordance with the guidelines on the website of the "Molecules" journal (https://www.mdpi.com/journal/molecules/instructions#preparation - Instructions for Authors - Research Manuscript Sections) that is: Introduction, Results, Discussion, Materials and Methods, Conclusions.
Dear Editor, Dear Reviewers,
We would like to submit a revised version of our manuscript entitled (new title – replacement of “Standardized” to “Water “as suggested by Reviewer #2):”The Effect of Different Water Extracts from Platycodon Grandiflorum on Selected Factors Associated with Pathogenesis of Chronic Bronchitis in Rats” for publication in “Molecules”. We appreciated the overall positive evaluation of our manuscript by the two reviewers and we would like to thank the reviewers for their efforts and constructive criticism that helped us to improve the quality of our manuscript. Please find below a point by point response to the issues raised by the reviewer. All changes in the manuscript are marked in yellow highlighting.
Reviewer #2
Comment 1.
In the title, the authors are using the term standardized extracts but it seems that it has an improper use at this We have to take in account that the chemical composition of a plant strongly depends on the environment, climate, harvesting period and so on. For this reason, the specification regarding the standardized plant extracts products should include identity, the quantity of declared and identified compounds, impurities if any (e.g. degradation products) and contaminants. Therefore, in a standardized extract, the compounds considered responsible for their pharmacological action should be provided each time at the same concentration. This article uses also roots of the field-cultivated plant, and the quantities of individual compounds were not provided. Therefore I would suggest removing the term “standardized” from the title and/or replace it with another word, more appropriate.
Response
Thank you for your suggestion. Indeed, according to the definition of "Standardized", all the extracts used should be subjected a detailed phytochemical analysis with the exact specification of the compounds for which the extract is standardized (e.g. the most characteristic, most abundant or having a specific pharmacological effect) and any other information that accurately describes the preparation of plant origin. It seems, however, that at this stage of knowledge of the composition of these extracts, such a procedure is impossible, without knowing all the properties of the tested extracts, which was partly the aim of the presented research. Hence, agreeing with the reviewer's remark, we decided to change this term to "Water", thus extending the title of the information on the type of extracts being the subject of the presented research.
Comment 2.
The use of plant name should be uniform during the manuscript - Platycodon Grandiflorum vs. Platycodi radix.
Response
As suggested, we unified the nomenclature throughout the text of the work by replacing "Platycodi radix" by using „ root of Platycodon glandiflorum” (for example: new lines 65, 276, 277, 286).
Comment 3.
Figure 5. Why * appears only for Contro+SO2?
Response
The * sign was used to define the occurrence of the difference for the value that was statistically significant in relation to the control group (it means rats that were not exposed to MBS = appropriate control negative study group) at the significance level p≤0.05. In fact, this * sign was only used to denote the difference to the value for positive control group (it means animals exposed to SO2 (= control + SO2)), thus emphasizing the effect of SO2. At the same time, other groups, i.e. those treated with SO2 and extract, were not marked with this sign intentionally. From the methodological point of view, it seems that in this way it is avoided to emphasize the effect of two factors at the same time, i.e. SO2 and the extract in comparison to the control negative group (control), because when analyzing such a phenomenon it is not possible to say clearly which of these two factors is important in relation to the control negative group (control). This method of marking significance was also used for the remaining results, i.e. in the appropriate tables.
Comment 4.
Try to avoid unclear or ambiguous formulations: line248- “other chemical substances”, line 263- “different cytokines”, line 267 – “a few cytokines”. Please be more specific.
Response
Thank you for your suggestion. Following your advice, we have revised these terms as follows:
- line 248 (new line 261) by deleting "other chemical substances" and now this sentence is changed as follows) ”Inflammatory reactions in the bronchial mucosa in respiratory tract diseases are controlled by set of interacting factors, e.g. cytokines.”
- line 263 (new line 274) by deleting “different” and now this sentence is changed as follows: “It is in accordance with publications by many authors, which clearly demonstrated that animals exposed to SO2 develop chronic inflammatory processes which are sustained by pro-inflammatory mediators, mainly cytokines”.
- line 267 (new line 278) by inserting “ Th1 and Th2 cytokines“ and now this sentence is changed as follows: “Extracts from the roots of PG markedly decreased the number of infiltrated inflammatory cells and the levels of Th1 and Th2 cytokines and chemokines compared with a control group and reduced ovalbumin-specific IgE levels in BALF fluid [9]”.
Comment 5.
Line 277 – typing mistake – NF-nB.
Response
Thank you for your suggestion. It was our typographic mistake. We changed this error to the correct wording of this factor in "NF-kB" (new line 288).
Comment 6.
Taking into account your affirmation from Lines 278-288 I think that you should further investigate also thinking from other perspectives. For example, what other compounds, not belonging to saponins and inulin were common to E1 and E2 extracts and might possess pharmacological effects? Also, you should try to find if you can correlate or explain the phenotypic differences with the differences in tissue expression patterns of TGFb –isoforms. The differences might be connected with the functional differences between TGF b isoforms. Can you find in vivo research articles that can support your findings?
Response
Thank you for your suggestion.
In trying to answer your questions, we have introduced new paragraphs in the text which we hope sheds some light on the issues raised in these.
- The answer to the question:„For example, what other compounds, not belonging to saponins and inulin were common to E1 and E2 extracts and might possess pharmacological effects?” is as follows:
New lines 299-314:
“The differences obtained in this study are not only related to different content of saponins and inulin (Tab. 1), but also to a slightly different qualitative composition of the examined extracts, as mentioned in the Results section (chapter 2.2. Qualitative analysis). This may be in line with the observation, that only one compound, No. 19, (platycodigenin gentobioside) is common to all extracts (Fig. 4). In addition to saponins (No. 15, 17, 19, 20, 21, 22, 23, 25, 26, 27, 30, 31, 33, 34, 35, 36, 37, 38, 39, 41, 42, 43, 44, 45, 46, 48), the presence of other groups of chemical compounds can be noticed (flavonoids: 6, 7, 9, 14, 24, 28, 29, 32, 40, 47; hydroxycinnamic acids: 1, 2, 3, 4, 5, 11, 16; lignans: 8, 10, saccharide: 12), which varied significantly among samples from the extracts (for example: flavonoids only in E1 extract). This is consistent with the knowledge of the possible contribution of these compounds present in PG to various pharmacological effects on inflammatory basis [58]. Hence, it cannot be ruled out that the observed differences in anti-inflammatory activity may have been caused by the presence of flavonoid glycosides, which were identified mainly in the E1 extract, whereas caffeic acid glycosides and 3-caffeoylquinic acid were characteristic only for E2 extract, and the isomer 5-caffeoylquinic acid and its glycoside were detected in the E3 extract (Tab. 3). Nevertheless, their possible significance should be supported by detailed quantification (content) of the above-mentioned compounds in the future”.
- The answer to the question:” Also, you should try to find if you can correlate or explain the phenotypic differences with the differences in tissue expression patterns of TGFb –isoforms. The differences might be connected with the functional differences between TGF b isoforms. Can you find in vivo research articles that can support your findings?” is as follows:
New lines 318-320:”The isoform TGF-β1 is responsible for fibrinogen activity in chronic bronchitis and plays a role in airway wall remodeling [2]. Moreover, this isoform produced the inhibitory action on immune cell differentiation (Th1 and Th2 cells and B cells) and cytokine production (IFN-? and IL-2) [59]”.
New lines 343-362:”The differences observed in the response of TGF-β isoforms to the action of SO2 and the administration of extracts may be related to a slightly different distribution and action of those in airways, since TGF-β1 colocalizes with extracellular matrix proteins, such as collagen, and interfaces between epithelial and mesenchymal cells, TGF-β2 is found in endodermal bronchiolar epithelium, whereas TGF-β3 is expressed in tracheal mesenchyme and the endodermal epithelial cells in bronchioles and mesodermal cells [60]. Moreover, the biological effects of different TGF-β isoforms depend on their availability, combination of two types of their receptors, and intracellular signaling pathways that they can induce [54]. For example, in the airways of humans with asthma, TGF-β1 levels are elevated as compared with normal control subjects, suggesting a role in the repair of injured asthmatic airways or the existence of a negative feedback loop controlling airway inflammation [60]. Similarly, an increase in TGF-β2 expression in asthmatic epithelium was shown, which correlates with an increase in the number of eosinophils and neutrophils in patients with severe and mild asthma [54]. There is relatively little information on TGF-β3 expression, although the available data suggests that there is no difference in TGF-β3 expression between asthmatic patients and control subjects [54], but in patients with severe COPD the lowering of TGF-β3 was found [55]. In conclusion, SO2, as a pro-inflammatory factor, induced changes in the expression of individual TGF-β isoforms in a similar way as the above-mentioned. However, the result of administration of the extracts to individual isoforms is not easily explained and will likely, as it was mentioned previously, requires in-depth research on the effect of not only saponins and inulin occurring in various amounts in the studied extracts, but also compounds from other chemical groups”.
- We found some information in additional cited works [new items -59 and 60].
Comment 7.
Please try to explain why in some cases the results were correlated with saponin or inulin concentration and in others no.
Response
We think that we have responded to this comment in some way in the first part of our reply to Comment 6.
Comment 8.
Please use the same writing format (font size) throughout the manuscript
Response
This note is of the same type as Reviewer # 1, Comment 2 and as suggested, we changed the font type and size used in the whole body text (Palatino linotype, size 10) - new lines 92-93, 128-130, 378-379, 401-413, respectively.
Comment 9.
Line 378-379 – you should be more specific.
Response
Thank you for your suggestion. However, due to the fact that this part of the text (new lines 417-418) is a kind of conclusion, it seems that such a generalizing nature of the sentence is justified at this point.
Comment 10.
Please put the materials and methods section after the introduction.
Response
Order of the chapters of this work was presented in accordance with the guidelines on the website of the "Molecules" journal (https://www.mdpi.com/journal/molecules/instructions#preparation - Instructions for Authors - Research Manuscript Sections) that is: Introduction, Results, Discussion, Materials and Methods, Conclusions.
Dear Editor, Dear Reviewers,
We would like to submit a revised version of our manuscript entitled (new title – replacement of “Standardized” to “Water “as suggested by Reviewer #2):”The Effect of Different Water Extracts from Platycodon Grandiflorum on Selected Factors Associated with Pathogenesis of Chronic Bronchitis in Rats” for publication in “Molecules”. We appreciated the overall positive evaluation of our manuscript by the two reviewers and we would like to thank the reviewers for their efforts and constructive criticism that helped us to improve the quality of our manuscript. Please find below a point by point response to the issues raised by the reviewer. All changes in the manuscript are marked in yellow highlighting.
Reviewer #2
Comment 1.
In the title, the authors are using the term standardized extracts but it seems that it has an improper use at this We have to take in account that the chemical composition of a plant strongly depends on the environment, climate, harvesting period and so on. For this reason, the specification regarding the standardized plant extracts products should include identity, the quantity of declared and identified compounds, impurities if any (e.g. degradation products) and contaminants. Therefore, in a standardized extract, the compounds considered responsible for their pharmacological action should be provided each time at the same concentration. This article uses also roots of the field-cultivated plant, and the quantities of individual compounds were not provided. Therefore I would suggest removing the term “standardized” from the title and/or replace it with another word, more appropriate.
Response
Thank you for your suggestion. Indeed, according to the definition of "Standardized", all the extracts used should be subjected a detailed phytochemical analysis with the exact specification of the compounds for which the extract is standardized (e.g. the most characteristic, most abundant or having a specific pharmacological effect) and any other information that accurately describes the preparation of plant origin. It seems, however, that at this stage of knowledge of the composition of these extracts, such a procedure is impossible, without knowing all the properties of the tested extracts, which was partly the aim of the presented research. Hence, agreeing with the reviewer's remark, we decided to change this term to "Water", thus extending the title of the information on the type of extracts being the subject of the presented research.
Comment 2.
The use of plant name should be uniform during the manuscript - Platycodon Grandiflorum vs. Platycodi radix.
Response
As suggested, we unified the nomenclature throughout the text of the work by replacing "Platycodi radix" by using „ root of Platycodon glandiflorum” (for example: new lines 65, 276, 277, 286).
Comment 3.
Figure 5. Why * appears only for Contro+SO2?
Response
The * sign was used to define the occurrence of the difference for the value that was statistically significant in relation to the control group (it means rats that were not exposed to MBS = appropriate control negative study group) at the significance level p≤0.05. In fact, this * sign was only used to denote the difference to the value for positive control group (it means animals exposed to SO2 (= control + SO2)), thus emphasizing the effect of SO2. At the same time, other groups, i.e. those treated with SO2 and extract, were not marked with this sign intentionally. From the methodological point of view, it seems that in this way it is avoided to emphasize the effect of two factors at the same time, i.e. SO2 and the extract in comparison to the control negative group (control), because when analyzing such a phenomenon it is not possible to say clearly which of these two factors is important in relation to the control negative group (control). This method of marking significance was also used for the remaining results, i.e. in the appropriate tables.
Comment 4.
Try to avoid unclear or ambiguous formulations: line248- “other chemical substances”, line 263- “different cytokines”, line 267 – “a few cytokines”. Please be more specific.
Response
Thank you for your suggestion. Following your advice, we have revised these terms as follows:
- line 248 (new line 261) by deleting "other chemical substances" and now this sentence is changed as follows) ”Inflammatory reactions in the bronchial mucosa in respiratory tract diseases are controlled by set of interacting factors, e.g. cytokines.”
- line 263 (new line 274) by deleting “different” and now this sentence is changed as follows: “It is in accordance with publications by many authors, which clearly demonstrated that animals exposed to SO2 develop chronic inflammatory processes which are sustained by pro-inflammatory mediators, mainly cytokines”.
- line 267 (new line 278) by inserting “ Th1 and Th2 cytokines“ and now this sentence is changed as follows: “Extracts from the roots of PG markedly decreased the number of infiltrated inflammatory cells and the levels of Th1 and Th2 cytokines and chemokines compared with a control group and reduced ovalbumin-specific IgE levels in BALF fluid [9]”.
Comment 5.
Line 277 – typing mistake – NF-nB.
Response
Thank you for your suggestion. It was our typographic mistake. We changed this error to the correct wording of this factor in "NF-kB" (new line 288).
Comment 6.
Taking into account your affirmation from Lines 278-288 I think that you should further investigate also thinking from other perspectives. For example, what other compounds, not belonging to saponins and inulin were common to E1 and E2 extracts and might possess pharmacological effects? Also, you should try to find if you can correlate or explain the phenotypic differences with the differences in tissue expression patterns of TGFb –isoforms. The differences might be connected with the functional differences between TGF b isoforms. Can you find in vivo research articles that can support your findings?
Response
Thank you for your suggestion.
In trying to answer your questions, we have introduced new paragraphs in the text which we hope sheds some light on the issues raised in these.
- The answer to the question:„For example, what other compounds, not belonging to saponins and inulin were common to E1 and E2 extracts and might possess pharmacological effects?” is as follows:
New lines 299-314:
“The differences obtained in this study are not only related to different content of saponins and inulin (Tab. 1), but also to a slightly different qualitative composition of the examined extracts, as mentioned in the Results section (chapter 2.2. Qualitative analysis). This may be in line with the observation, that only one compound, No. 19, (platycodigenin gentobioside) is common to all extracts (Fig. 4). In addition to saponins (No. 15, 17, 19, 20, 21, 22, 23, 25, 26, 27, 30, 31, 33, 34, 35, 36, 37, 38, 39, 41, 42, 43, 44, 45, 46, 48), the presence of other groups of chemical compounds can be noticed (flavonoids: 6, 7, 9, 14, 24, 28, 29, 32, 40, 47; hydroxycinnamic acids: 1, 2, 3, 4, 5, 11, 16; lignans: 8, 10, saccharide: 12), which varied significantly among samples from the extracts (for example: flavonoids only in E1 extract). This is consistent with the knowledge of the possible contribution of these compounds present in PG to various pharmacological effects on inflammatory basis [58]. Hence, it cannot be ruled out that the observed differences in anti-inflammatory activity may have been caused by the presence of flavonoid glycosides, which were identified mainly in the E1 extract, whereas caffeic acid glycosides and 3-caffeoylquinic acid were characteristic only for E2 extract, and the isomer 5-caffeoylquinic acid and its glycoside were detected in the E3 extract (Tab. 3). Nevertheless, their possible significance should be supported by detailed quantification (content) of the above-mentioned compounds in the future”.
- The answer to the question:” Also, you should try to find if you can correlate or explain the phenotypic differences with the differences in tissue expression patterns of TGFb –isoforms. The differences might be connected with the functional differences between TGF b isoforms. Can you find in vivo research articles that can support your findings?” is as follows:
New lines 318-320:”The isoform TGF-β1 is responsible for fibrinogen activity in chronic bronchitis and plays a role in airway wall remodeling [2]. Moreover, this isoform produced the inhibitory action on immune cell differentiation (Th1 and Th2 cells and B cells) and cytokine production (IFN-? and IL-2) [59]”.
New lines 343-362:”The differences observed in the response of TGF-β isoforms to the action of SO2 and the administration of extracts may be related to a slightly different distribution and action of those in airways, since TGF-β1 colocalizes with extracellular matrix proteins, such as collagen, and interfaces between epithelial and mesenchymal cells, TGF-β2 is found in endodermal bronchiolar epithelium, whereas TGF-β3 is expressed in tracheal mesenchyme and the endodermal epithelial cells in bronchioles and mesodermal cells [60]. Moreover, the biological effects of different TGF-β isoforms depend on their availability, combination of two types of their receptors, and intracellular signaling pathways that they can induce [54]. For example, in the airways of humans with asthma, TGF-β1 levels are elevated as compared with normal control subjects, suggesting a role in the repair of injured asthmatic airways or the existence of a negative feedback loop controlling airway inflammation [60]. Similarly, an increase in TGF-β2 expression in asthmatic epithelium was shown, which correlates with an increase in the number of eosinophils and neutrophils in patients with severe and mild asthma [54]. There is relatively little information on TGF-β3 expression, although the available data suggests that there is no difference in TGF-β3 expression between asthmatic patients and control subjects [54], but in patients with severe COPD the lowering of TGF-β3 was found [55]. In conclusion, SO2, as a pro-inflammatory factor, induced changes in the expression of individual TGF-β isoforms in a similar way as the above-mentioned. However, the result of administration of the extracts to individual isoforms is not easily explained and will likely, as it was mentioned previously, requires in-depth research on the effect of not only saponins and inulin occurring in various amounts in the studied extracts, but also compounds from other chemical groups”.
- We found some information in additional cited works [new items -59 and 60].
Comment 7.
Please try to explain why in some cases the results were correlated with saponin or inulin concentration and in others no.
Response
We think that we have responded to this comment in some way in the first part of our reply to Comment 6.
Comment 8.
Please use the same writing format (font size) throughout the manuscript
Response
This note is of the same type as Reviewer # 1, Comment 2 and as suggested, we changed the font type and size used in the whole body text (Palatino linotype, size 10) - new lines 92-93, 128-130, 378-379, 401-413, respectively.
Comment 9.
Line 378-379 – you should be more specific.
Response
Thank you for your suggestion. However, due to the fact that this part of the text (new lines 417-418) is a kind of conclusion, it seems that such a generalizing nature of the sentence is justified at this point.
Comment 10.
Please put the materials and methods section after the introduction.
Response
Order of the chapters of this work was presented in accordance with the guidelines on the website of the "Molecules" journal (https://www.mdpi.com/journal/molecules/instructions#preparation - Instructions for Authors - Research Manuscript Sections) that is: Introduction, Results, Discussion, Materials and Methods, Conclusions.
